


# Scale-dependency of extreme precipitation processes in regional climate simulations of the greater Alpine region

Alberto Caldas-Alvarez[1], Hendrik Feldmann[1], Etor Lucio-Eceiza[2,3], and Joaquim Ginete Pinto[1]

[1]Institute of Meteorology and Climate Research (IMK-TRO), Karlsruhe Institute of Technology (KIT), Karlsruhe, 76131, Germany.
[2]Institute of Meteorology, Freie Universität Berlin (FUB), Berlin, 14195, Germany.
[3]Deutsches Klimarechenzentrum (DKRZ, German Climate Computing Center), Hamburg, 20146, Germany.

*Correspondence to*: Alberto Caldas-Alvarez (alberto.caldas-alvarez@kit.edu)

**Abstract.** Heavy Precipitation Events (HPEs) are a challenging atmospheric phenomenon with a high impact on human lives and infrastructures. The achievement of high-resolution simulations for Convection Permitting Modelling (CPM) has brought relevant advancements in the representation of HPEs in climate simulations compared to coarser resolution Regional Climate Models (RCM). However, further insight is needed on the scale-dependency of mesoscale precipitation processes. In this study, we aim at evaluating reanalysis-driven climate simulations of the greater Alpine area in recent climate conditions and assessing the scale-dependency of thermodynamical processes influencing extreme precipitation. We evaluate COSMO-CLM simulations of the period 1971–2015, at resolutions of 25 km (RCM) and 3 km (CPM) downscaled from ERA-40 and ERA-interim. We validate our simulations against high-resolution observations (EOBS, HYRAS, MSWEP, and UWYO). In the methodology, we present a revisited version of the Precipitation Severity Index (PSI) useful for extremes detection. Furthermore, we obtain the main modes of precipitation variance and synoptic Weather Types (WTs) associated with extreme precipitation using Principal Component Analysis (PCA). PCA is also used to derive composites of model variables associated with the thermodynamical processes of heavy precipitation. The results indicate a good detection capability of the PSI for precipitation extremes. We identified four WTs as precursors of extreme precipitation in winter, associated with stationary fronts or a zonal flow regimes. In summer, 5 WTs bring heavy precipitation, associated with upper-level elongated troughs over western Europe, sometimes evolving into cut-off lows, or by winter-like situations of strong zonal circulation. The model evaluation showed that CPM (3 km) represents higher precipitation intensities, better rank correlation, better hit rates for extremes detection, and an improved representation of heavy precipitation amount and structure for selected HPEs compared to RCM (25 km). CPM overestimates grid point precipitation rates especially over elevated terrain fostered by the scale-dependency of convective dynamic processes such as stronger updrafts and more triggering of convective cells. However, at low altitudes, precipitation differences due to resolution are explained through the scale-dependency of thermodynamic variables, where the largest impact is caused by differences in surface moisture up to 1 g kg$^{-1}$. These differences show a predominant north-south gradient where locations north of the Alps show larger (lower) surface moisture and precipitation in CPM (RCM) and locations south of the Alps show larger (lower) humidity and precipitation in RCM (CPM). The humidity differences are caused by an uneven partition of latent and sensible heat fluxes between RCM and CPM. RCM simulates larger



emissions of latent heat flux over the Sea (25 W m$^{-2}$ more), and CPM emits larger latent heat over land (15 W m$^{-2}$ more). In

turn, RCM emits larger surface sensible heat fluxes over land (30 W m$^{-2}$ more), showing a warmer surface (0.5 to 1°C) than CPM. These results provide evidence that CPM is a powerful tool for obtaining accurate high-resolution climate information also pointing at the different scale-dependency of dynamic and thermodynamical precipitation processes at high and low terrain.

## 1 Introduction


Heavy Precipitation Events (HPEs) are one of the main natural hazards affecting Central Europe, often causing tremendous damages and casualties (Alfieri et al., 2016; Khodayar et al., 2021; Ranasinghe et al., 2021). The recent event affecting western Germany, Belgium, Netherlands, and Luxembourg in July 2021 caused over 170 casualties and losses above 10 bill. Euro (Schäfer et al., 2021). In a warming climate, the occurrence and intensity of such events is projected to increase as assessed in

Chapter 8 of the Intergovernmental Panel on Climate Change (IPCC) and previous publications (Douville et al., 2021; Pichelli et al., 2021), due to the intensification of the hydrological cycle (Rajcack and Schär, 2013; Ban et al., 2018). Such events may occur both during winter and summer fostered by Deep Moist Convection (DMC), a large vertical transport of precipitating air masses (Emanuel; 1994). In Winter, extreme precipitation typically occurs under strong synoptic forcing (Keil et al., 2020), caused by the large-scale advection of positive vorticity in cold upper-level layers (Holton, 2013). In summer, DMC is often

triggered by favourable local and mesoscale conditions close to the surface, including an energetic and moist low-level and a triggering mechanism (Doswell, 1996). When these conditions coincide with the arrival of a mesoscale low-pressure system, highly damaging precipitation is likely to occur. Summer HPEs are hence characterized by strong, localized, and short convective showers that have a high risk of causing flash flooding (Doswell et al., 1996; Khodayar et al., 2021).

Understanding heavy precipitation processes, their variability and trends at the climate scale is thus needed to provide better

prevention and adaptation strategies. Considering modelling approaches, dynamical downscaling with regional climate models (RCM) has proven to be an important tool towards this end, even though convection is parameterised (e.g. Jacob et al.,2013). Recently, the development Convection-Permitting Models (CPMs) led to a further step forward (Coppola et al., 2018; Prein et al., 2020; Lucas-Picher et al., 2021). The added value of CPM lies primarily in the explicit representation of convection, provided a horizontal resolution higher than ca. 3 km is attained. Also improved is the representation of the model's land type,

use and elevation (Prein et al., 2015; Heim et al., 2020). These advancements led to improvements in representing the precipitation's diurnal cycle (Kendon et al., 2012; Berthou et al., 2018; Ban et al., 2021); its structure, intensity, frequency, and duration (Berthou et al., 2019; Berg et al., 2019); its sub-hourly rates (Meredith et al., 2020); and orographic triggering (Ban et al., 2018). These improvements are consistent over the main modelling regions worldwide. However, not all problems are solved, since CPMs have also shown relevant wet biases, inducing an overestimation of extreme intensities (Kendon et al.,





2012). CPM uncertainties arise from shortcomings in the physical parameterizations, deficiencies in the representation of the
     initial conditions and the lack of sufficient high-resolution observations for validation (Lucas-Picher et al., 2021).

Particularly relevant for the improvement of heavy precipitation in CPM is the better representation of the dynamical processes
of DMC, especially when convection is triggered close to the surface (Bui et al., 2018). In fact, several studies have shown
that CPMs induce stronger updraughts and thus larger precipitation than coarse resolution models (Meredith et al., 2015a;

Meredith et al., 2015b). This is also observed in Numerical Weather Prediction (NWP) simulations (Barthlott and Hoose, 2015;
     Panosetti et al., 2018). When convection occurs over an area of complex orography, the finer representation of the mountains
     in CPM increases the triggering of convection (Langhans et al., 2012; Vanden Broucke et al., 2018; Heim et al., 2018), leading
     to a better agreement with radar observations (Purr et al., 2019).

Regarding the scale-dependency of thermodynamical precipitation processes, previous papers argued that CPMs improve the

simulation of surface temperature (Ban et al., 2014; Prein et al., 2015; Hackenbruch et al., 2016), due to a better representation
     of the orography, the precipitation's location and the cloud coverage (Lucas-Picher et al., 2021). Hohenegger et al., (2009)
     showed that using CPM favours a negative soil-precipitation feedback (more rain under dry soil conditions), as opposite to
     convection-parameterized RCM (25 km), which show a positive feedback (more rain under wet soil conditions). The negative
     bias in CPM is due to stronger thermals in CPMs given dry soil conditions, thus initiating convection (Hohenegger et al.,

2009). Moisture biases also affect the development of extreme precipitation where a wet bias was found for established RCM
     models (Lin et al., 2018; Li et al., 2020), as well as in CPM simulations (Risanto et al., 2019; Bastin et al., 2019; Caldas-
     Alvarez and Khodayar, 2020; Li et al., 2020). However, how both RCM and CPM deal with the moisture excess still is an
     open question. Regarding atmospheric instability Li et al., (2020), found larger Convective Available Potential Energy (CAPE)
     during the afternoon in CPM, which was correctly converted to larger precipitation at the Tibetan Plateau. However, more

knowledge on the interplay of moisture and instability in CPM at the European continent is needed. Finally, the scale
     dependency of other variables of interest for convective development such as Equivalent Potential Temperature at 850 hPa
     ($\theta_e^{850}$), has been seldom investigated.

The aim of this work is to evaluate reanalysis-driven RCM (25 km) and CPM (3 km) decadal long simulations of the greater
Alpine area and assess the scale-dependency of thermodynamical processes influencing extreme precipitation, focusing on the

period 1971 to 2015. Particular research questions that we would like to answer are, how can we better detect precipitation
     extremes? can we assess the large-scale variability and main synoptic patterns associated with extreme events? How is heavy
     precipitation representation affected by the use of CPM? How much are processes such as energy and moisture fluxes,
     instability or surface fluxes affected by CPM?

In Sect. 2 we introduce the dataset and methods employed; in Sect. 3 we present the main synoptic weather types bringing

extreme precipitation; in Sect. 4 we evaluate extreme precipitation intensity and occurrence in the climate simulations; in Sect.
     5 we validate precipitation, humidity, and temperature fields of selected extreme precipitation events; in Sect. 6 we assess the



scale dependency of moisture and energy fluxes, instability, and soil-atmosphere fluxes. Finally in Sect. 7 we present our conclusions.

## 2 Data and methods

### 2.1 Observational datasets

We use observations from different sources for validation and comparison of the climate simulations (Tab. 1). The Ensembles OBSservations (EOBS) gridded precipitation and relative humidity at the surface ($hurs$) products are provided by the European Climate Assessment & Dataset (ECAD) centre at 0.1° (ca. 11 km) and 0.25° (ca. 25 km) spatial resolutions for the period 1950-2020. We use v.22.0e (Dec 2020), where EOBS-25km employs a 100-member ensemble created through stochastic simulations based on interpolated station data from national institutions including 9000 rain gauges (Cornes et al., 2018). The 25 km resolution and daily aggregations are used for validation of precipitation simulations for 1971-2015 to match the resolution of the evaluated RCM simulation (Fig.1). EOBS-25km has been widely used in previous literature for validation purposes (e.g., Tramblay et al., 2019; Bandhauer et al., 2021) and has been shown to have low median absolute biases with respect to other regional European precipitation products such as CARPATCLIM or Spain02 (Cornes et al., 2018).

The HYdrologische RASterdatensaetze (HYRAS) gridded precipitation dataset, provided by the German Weather Service (DWD) is available at 1 km (ca. 0.01°), 5 km (ca. 0.05°) and daily resolution. HYRAS covers Germany and neighbouring catchments in parts of Switzerland, Austria, the Netherlands, France, Belgium, and Poland (Fig.1). The version v2 covers the period 1951-2015 and was derived using multiple linear regression and inverse distance weighting interpolation of 6200 rain gauges considering the orography (Rauthe et al., 2013, Razafimaharo et al., 2020). HYRAS has a very high quality in extreme precipitation observations compared to the European Centre for Medium-Range Weather Forecasts REanalyes, 5th generation (ERA-5) and EOBS-25km (Hu et al., 2020). Its high-resolution enables a good representation of local scale features, outperforming the coarse resolution of EOBS-25km. However , it is only available over Germany and nearby catchments.

The Multi-Source Weighted-Ensemble Precipitation (MSWEP) is a gridded precipitation product provided by GloH2O (http://www.gloh2o.org/) at 0.1° (ca. 11 km) spatial resolution and 3-hourly temporal resolution for the period 1979-2020 with global coverage. We use version v.2.2.0. MSWEP obtained through weighted interpolation of different observations to a common grid. It merges data from rain Gauge observations from Climate Prediction Center (CPC) unified and Global Precipitation Climatology Centre (GPCC), satellite observations from the CPC MORPHing product (CMORPH), Global Satellite Mapping Precipitation Moving Vector with Kalman (GSMaP-MVK) and Tropical Rainfall Measuring Mission Multi-Satellite Precipitation Analysis (TMPA) 3B42, as well as two reanalyses datasets ERA-interim and Japanese Reanalyses JRA-55 (Beck et a., 2019). MSWEP has a high median correlation up to 0.67 against stations, compared to CMORPH (0.44) and TMPA-3B42 (0.59) (Beck et al., 2017). We use the MSWEP product to profit from its high accuracy and to overcome the limitations of HYRAS-5km (limited spatial coverage) and EOBS-25km (land-only).





The radiosonde data archived by the University of Wyoming (UWYO) are used to validate the RCM and CPM humidity and temperature profiles. The stations are located close to large European cities, with an average distance of 250 km. The temporal
resolution ranges between 6 h, 12 h and 24 h, The provided information includes height, atmospheric pressure, temperature, and dew point temperature on ca. 30 levels between 1000 and 300 hPa, as well as integrated convective indices. The UWYO soundings are often used as reference for validation studies (e.g., Ciesileski et al., 2014; Yang et al., 2020).

**2.2 Methodology: Comparison performance RCM vs CPM**

We systematically compare regional climate simulations with the Consortium for Small-scale MOdeling (COSMO) model in
CLimate Mode (CCLM), of the recent climate at two model resolutions; 25 km, hereafter named RCM and 3 km, named CPM. In the latter, deep convection is explicitly resolved. The RCM simulation (Tab.2) covers the period 1961-2018, has a nominal resolution of 0.22° (ca. 25 km), a 6-hourly output, and was performed within the scope of the finalized project MiKlip phase II (Feldmann et al., 2019). This simulation was performed for the Euro-CORDEX domain (Jacob et al., 2014) and thus covers the European continent and vast areas of the North Atlantic and the Mediterranean (Fig.1). The RCM simulation is forced by
ERA-40 Reanalysis (Uppala et al., 2005) until 1979 and by ERA-interim (Dee et al., 2011) afterwards, due to the availability these data sets. The physical packages and model settings are consistent throughout the whole period 1961-2018. The most relevant model settings are summarized in Tab. 2 and in Feldmann et al., (2019). CPM is built up from two separate simulations (Tab. 2). The first one is the KLIWA-2.8km simulation, which spans the period 1971 to 1999, is forced by ERA-40, has a 0.025° resolution (ca. 2.8 km), a 6-hourly output, covers the region of southern Germany and Switzerland (Fig.1b) and was
performed for the "Klimaveränderung und Wasserwirtschaft" (KLIWA) project (Hackenbruch et al., 2016). The second simulation ALP-3 simulation spans from 2000 to 2016, is forced by ERA-interim, has a resolution of 0.0267° (ca. 3km), a 6-hourly output, covers the greater Alpine area (Fig.1b), is part of the Flagship Pilot Study (FPS) Convection program (Coppola et al., 2018) funded by the World Climate Research Programme (WCRP). Both CPM simulations have the same model dynamics, the same integration time step (20 seconds), the same grid type (Arkawa-C), the same output frequency (6-hourly)
and the same active parameterizations. The physical parametrizations are the same as in RCM, except for the convection parametrization which is restricted to shallow convection (Schättler et al., 2016). However, KLIWA-2.8km and ALP-3 differ in the number of vertical levels (30 vs 40), the spatial resolution (2.8 km vs 3 km), the forcing data (ERA-40 vs ERA-interim), and the simulation domains (Fig. 1). In spite of these small inconsistencies, we combine both CPM simulations to attain a sufficiently large investigation period for comparison with the RCM simulation and observational datasets.

Two areas are investigated in our study. The first, denominated southern Germany (SGer, Fig.1), is used to provide a validation of precipitation for the complete period 1971-2015 (Sect. 4). This area and period were selected as it fulfil the requirements of all data sets (availability, coverage, time span). The second area, ALP-3 (Fig. 1), is used for validation of HPEs and the in-depth analysis of the scale-dependency of precipitation-related variables for the period 2000-2015 (Section 6).





### 2.3 Analytical methods

**2.3.1 The Precipitation Severity Index (PSI)**

We employ the PSI to detect the extreme events including three different, but intertwined aspects of extreme precipitation: grid-point intensity, spatial extent of affected area and time persistence. The PSI is adapted from the Storm Severity Index (SSI; Leckebusch et al., 2008; Pinto et al., 2012) and further developments by Piper et al., (2016). It is defined as follows:

$$PSI_T = \frac{1}{(1+d) \cdot A} \sum_{i=1}^{N} \sum_{j=1}^{M} \sum_{t=T-d}^{T} \frac{RR_{ijt}}{RR_{perc_{ij}}} \cdot (\Delta x)^2 \cdot \prod_{\tau=t}^{T} I\left(RR_{ij\tau}, RR_{perc_{ij}}\right)$$ [1]


$$I\left(RR_{ij\tau}, RR_{perc_{ij}}\right) = \begin{cases} 0 \ if \ RR_{ij\tau} \leq RR_{80_{ij}} \\ \\ 1 \ if \ RR_{ij\tau} > RR_{80_{ij}} \end{cases}$$

The PSI values at a certain time step T ($PSI_T$) depends on the ratio between grid point daily precipitation ($RR_{ijt}$) and a

percentile of the climatology ($RR_{perc_{ij}}$). We set this threshold to be the 80-percentile to ensure that only precipitation events with high grid-point intensity are considered. We thus neglect grid points whose precipitation is lower than the set threshold one for day T ($RR_{ij\tau} \leq RR_{perc_{ij}}$), by means of the function $I\left(RR_{ij\tau}, RR_{perc_{ij}}\right)$. We consider precipitation persistence through the sum over time ($t$). The ratios at each grid point for day T and the previous $d$ days ($d = 2$ in our case) are added for the PSI calculation, provided precipitation was continuous and larger than $RR_{perc_{ij}}$ at that same grid point $i, j$. To consider the size of

each grid cell we multiply by the area of one grid cell ($(\Delta x)^2$). Finally, the ratios are summed over the spatial extent ($NxM$) along directions $i$ and j. The daily PSI value is normalized to the area of the simulation domain $A = N \cdot M \cdot (\Delta x)^2$ multiplied by $(1 + d)$ to consider the addition of grid points with persistent precipitation. Prior to the PSI calculation, we include a correction for latitude stretching of the grid as $sqrt(cos(lat))$ following (North et al., 1982)

To assess the performance of the PSI in detecting HPEs, we analyse in detail the exceptional heavy precipitation season of

spring 2007 in southern Germany and the Alps (Schacher et al., 2007). Figure 2 shows the temporal evolution of daily PSI values at SGer for different settings of the percentile in $RR_{perc_{ij}}$ and of the persistence parameter ($d$). Additionally, the temporal evolution of the field sum ($fldsum$) is shown. The PSI detects the six HPEs affecting SGer during the season, namely, on the 08 May, 14 May, 28 May, 14 Jun, 22 Jun, and 25 Jun (Fig. 2). For a fixed value of the percentile ($perc = 80$), the impact of considering persistent precipitation during the two days prior to the analysed date (perc-80-days-2; black) is to

reduce the PSI values and to shift the peaks to the last day of the event (see events 1, 3, and 5), compared to considering no days of persistent precipitation (perc-80-days-0; dark grey). On the other hand, for a fixed value of the persistence parameter ($d = 2$), increasing the percentile (perc-95-days-2; light grey) does not affect the timing of the temporal evolution, but reduces the PSI values and neglects episodes of stratiform rain (11-Jun). This analysis provides confidences that the selected parameters





$perc = 80$, and $d = 2$, are suitable for summer and winter extremes detection, not to neglect all stratiform precipitation events
and to identify cases where precipitation occurs over already very wet areas.

The comparison against $fldsum$ (dashed line), shows high spearman's rank correlations of the PSI, between 0.97 and 0.86, depending on the $prec$ and $d$ parameters. This implies that the PSI detects events ranked by highest maximum areal precipitation covering the functionality of $fldsum$. However, the PSI (perc-80-days-2; black) can help neglect weaker episodes such as the 11-Jun precipitation. Fig. 2 shows the 90-percentile of the 1971-2015 climatology over SGer for the PSI
(black horizontal line), and the $fldsum$ (dashed horizontal line) as a plausible threshold to detect extremes. Whereas the stratiform event of 11-Jun belongs to the 90-percentile of $fldsum$, it is neglected by the PSI, that only identifies events of very high grid-point intensity.

We conclude that the PSI is a suitable index for extreme's detection that adds value with respect to a simpler field sum by favouring the detection of events with large grid point intensities and persistent heavy precipitation.

## 2.3.2 Principal Component Analysis

Principal Component Analysis (PCA) is a method to reduce the dimensionality of a data set, by transforming it to a new coordinate system of variables called Principal Components (PCs; Joliffe, 2002). The functions that allow the transformation from the original set to the PCs space are called Empirical Orthogonal Functions (EOFs). The transformation is performed in such a way that the explained variance is concentrated in a small number of the new variables. By construction, the leading
EOF1 has the largest explained variance, followed by EOF2, and so on. In this paper, we investigate the PCs and EOFs of 500 hPa geopotential height fields (Sect. 3.1) and daily precipitation (Sect. 5). Similarly to Ulbrich et al., (1999), we obtain EOFs representing the spatial patterns of the target variable, that account for the main modes of variance. On the other hand, the PCs are time series which provide the information of the loading of each EOF at a specific time step.

Given that the explained variance is now concentrated in a small number of variables, it is important to discern how many
EOFs should be retained. With this aim, we use a method of parallel analysis based on the randomization of eigenvalues named the random-λ rule (Peres-Neto, 2005). The procedure is as follows, 1) a random data array is created with the same dimensions as the data array under study, 2) PCA is applied on the random array, 3) steps 1 and 2 are repeated up to 1000 times, retaining at the eigenvalues showing a significance over 95 % (alpha= 0.05). 4). If the original eigenvalues exceed the critical values from the random data, then we reject the null-hypothesis (Peres-Neto, 2005). The random-λ rule is more suitable than other
methods of parallel analysis such as the N-rule (Preisendorfer and Mobley, 1988) since it does not assume a normal distribution for the array of random values and thus works better for variables such as precipitation.





### 2.3.3 Validation metric Fractions Skill Score

The Fractions Skill Score (FSS) provides an estimation of the model's skill in representing the fraction of surface affected (or not) by heavy precipitation (Skok and Roberts, 2016). A perfect forecast has thus an FSS of 1. A simulation with no skill has

an FSS of 0. In this work, we set a threshold of 40 mm d$^{-1}$ to define structures affected by heavy precipitation. The threshold is in the range values implemented by Roberts and Lean (2008) for simulations of spring convective rain over southern England. We select this threshold to be able to identify clear precipitation structures otherwise masked by the choice of a too large or too low threshold analogously to Caldas-Alvarez et al., (2021). Equation 2 defines the FSS following Roberts and Lean (2008).

$$FSS = 1 - \frac{\frac{1}{M}\sum_{i=1}^{M}(f_{mod}-f_{obs})^2}{\frac{1}{M}(\sum_{i=1}^{M}f_{mod}^2 + \sum_{i=1}^{M}f_{obs}^2)} \qquad [2]$$

The fractions of surface affected by heavy precipitation are represented by $f_{obs}$ and $f_{mod}$, for the observations and the model, respectively. Both are calculated as the number of grid points affected by precipitation over the defined threshold (40 mm d$^{-1}$) divided by the total number of grid points of a domain. FSS is computed as the ratio of the sums of fraction differences for M sub-boxes within the investigation domain. These M sub boxes are defined as sub-domains around M grid points with N near

neighbours. N in our case is 12 since most of the events we validate have shown a skill larger than the target skill defined as $FSS_{target} = 0.5 + f_{obs}/2$ for $N = 12$. For detailed explanation, refer to Roberts and Lean (2008), Skok et al., (2016), and Caldas-Alvarez et al., (2021).

### 3 Synoptic weather types

We obtain the predominant large-scale situations associated with heavy precipitation applying PCA. We analyse the EOFs of

geopotential height at 500 hPa, based on the RCM simulation, for the period 1971-2015. We select dates of heavy precipitation in the 98-percentile of severity (PSI) in the HYRAS-5km data set over the investigation region SGer (Fig. 1). Figures 3 and 4 provide, respectively, the dominating weather types of extreme precipitation for summer (MAMJJA) and winter (SONDJF). The comparison against the CPM is not shown here since only negligible differences exist with respect to RCM. This is because the boundary conditions from the forcing reanalyses (ERA) strongly determine the large-scale features under play (Prein et al.,

240  2015).

In winter, four synoptic patterns of 500 hPa geopotential height suffice to explain the natural variability, flowing the random-λ rule with a 95% significance in the t-test (Peres-Neto et al., 2005). They account for 74% of the heavy precipitation episodes. The first mode, representing 29 % of the events, is characterized by wave trains of low pressure associated with northerly incursions of polar air (Fig. 3). The synoptical situation is analogue to the Stationary Fronts (STF) category proposed by Stucki

et al., (2012). In this situation, heavy precipitation over the Alps is associated with strong upper-level lifting over northern



Italy and large southwesterly advection of moisture from the Mediterranean. Historical cases belonging to this category, as identified by the PCA, are the second phase of the 23-31 October storms in 1998 (Fuchs et al., 1998) or the late November events in 2015 (Tab. 3, https://www.wetter.de/cms/so-war-das-wetter-im-november-2015-2566771.html), for instance. The second mode, accounting for 22 % of the events, shows strong north-south gradients of the 500 hPa height and fast zonal circulations (Fig. 3). This synoptic pattern has been identified as a Zonal Flow (ZOW; Stucki et al., 2012) or as a narrow and elongated streamer (Massacand et al., 1998). The zonal circulation favours moisture advection from the Atlantic and can produce large precipitation in non-convective environments (Stucki et al., 2012). The 29 December 2001 event belongs to this precipitation mode, for instance. The third and four modes account for 12 % and 11 % of precipitation episodes, respectively and show similarities with the 500 hPa geopotential heights of the second mode (Fig. 3). However, the third synoptic pattern shows a weaker Azores high, favouring the advection of Atlantic moisture with a southwesterly component. The fourth mode, for its part, shows a weaker polar low, which favours the development of anti-cyclonic circulation (Fig. 3).

In summer, five synoptic patterns of 500 hPa geopotential height are discernible from random noise (Peres-Neto et al., 2005), accounting for 77 % of the events. The first mode, corresponding to 27% of the considered dates, shows an extended upper-level trough from the British Isles down to southern France (Fig. 4). This configuration shows elements of an Elongated Cut-Off (ECO) and of CAnarian Troughs (CAT; Stucki et al., (2012). In such situations upper-level lifting occurs east of the trough together with southerly moisture advection either from the southwest or the southeast, respectively. Such situation occurred for instance during the first stages of the large central European flooding of early June 2013 (Klemen et al., 2016). If a blocking situation occurs, for instance Omega blocking, the persistence of precipitation is enhanced and can lead to recurrent events (Kautz et al., 2021) at the eastern flank of the ECO or CAT . The second summer precipitation mode (Fig. 4), accounting for 19% of the events, presents a similar pattern to the third and four modes of winter precipitation (Fig. 3) with the characteristic strong zonal flow from the Atlantic. Examples of this synoptic configuration are the March 1988 events flooding the Rhein river (southern western Germany; Prellberg and Fell, 1989) or the 15 June 2007 events affecting southern Germany (https://www.wetteronline.de/extremwetter/schwere-gewitter-und-starkregen-schaeden-durch-tief-quintus-2007-06-15-tq).

The third precipitation mode, explaining 12 % of the analysed days (Fig. 4), shows similarly to the first mode, an ECO, however, with an eastward shifting of the Azores ridge and the possibility of evolving to a Pivoting Cut-Off Low (PCO; Stucki et al., 2012). If the PCO finally realizes and reaches the Mediterranean it is accompanied by a cyclonic flow brining moisture originating at the Balkan region. This has been demonstrated to be the case for the second phase of the June 2013 flooding (Klemen et al., 2016). The fourth summer precipitation mode (Fig. 4), accounts for 11% of the considered episodes and represents situations of northeasterly development of the upper-level trough. The low pressure evolves into a CAT situation inducing a southwesterly moist inflow to the Alpine region (Stucki et al., 2012). The 08 July 2004 floods in Baden-Wuerttemberg (southwestern Germany; http://contourmap.internet-box.ch/app/okerbernhard/presse2.htm) are a good example of such situation. The fifth precipitation mode, 8 % of the events, shows an STF pattern, similarly to the first winter precipitation mode (Fig. 3). Such a configuration was present during the Rhein-Necker flooding (western Germany) in June



2005          (https://www.rnz.de/nachrichten/metropolregion_artikel,-unwetter-folgen-in-mannheim-besonders-viele-

gebaeudeschaeden-durch-regen-_arid,482078.html).

## 4 Evaluation of extreme precipitation

After identifying the synoptic situations responsible for heavy precipitation, we evaluate Consortium for SMall Scale Modelling OSMO in Climate Mode (COSMO-CLM) simulations of the recent climate (Tab. 2) in terms of probability, intensity, and detection capability of extreme precipitation comparing the model results against observations.

Figure 5 shows box-and-whiskers distributions of daily precipitation larger than 1 mm d$^{-1}$ between 1971 and 2015 over SGer (Fig. 1). RCM (blue) and CPM (red) represent similar mean precipitation for the period (ca. 7 mm d$^{-1}$) but that CPM has a tail shifted towards the extremes. The 99-percentile for CPM (red, vertical bar) reaches 40 mm d$^{-1}$, whereas RCM (blue, vertical bars) is below 35 mm d$^{-1}$. The same occurs for maximum grid point precipitation. For CPM it reaches 360 mm d$^{-1}$ for CPM, whereas RCM has a 99-percentile of 250 mm d$^{-1}$. The ability of CPM to represent larger precipitation rates agrees with previous

literature (Ban et al., 2014; Prein et al., 2015; Fosser et al., 2014; ), which has been related to the enhanced intensities over orographic terrain (Langhans et al., 2012; Vanden Broucke et al., 2018; Ban et al., 2021). The comparison against HYRAS-5km (black), shows a good agreement by RCM and CPM for values between 1 mm d$^{-1}$ and 10 mm d$^{-1}$. However, CPM (red) overestimates extreme precipitation for grid point maxima. This does not imply a worse performance by CPM for event representation since despite local grid point overestimations by CPM previous studies found robust improvements in the

representation of the diurnal cycle, and the structure compared to RCM (Kendon et al., 2012; Lin et al., 2018). Furthermore, CPMs have shown systematically better results when validating events, once aggregated in space and time (Chan et al., 2012; Ban et al., 2018). Finally, Fig. 5 also shows the agreement between HYRAS-5km and EOBS-25km with however a better capacity of HYRAS-5km to represent large precipitation intensities, shown by the 99-percentile (vertical bar) and the grid point maxima (vertical line).

To further assess the performance of COSMO-CLM in representing precipitation extremes we analyse the detection capabilities of RCM (blue circles) and CPM (red dots) by means of a dot diagram, showing the 500 most severe events detected with the PSI in the period 1971-2015 over SGer (Fig. 6). We use HYRAS-5km (black circles and EOBS-25km (grey squares) as reference. CPM (red dots) showed a higher spearman's rank correlation (0.48) than RCM (blue circles; 0.41) as shown in the legend of Fig. 6. The same applies to hit rate with values of 47.2 % for CPM and 45.88 % for RCM (not shown). The

improvement shown by CPM with respect to RCM is shows some added value of high-resolution in representing extreme precipitation events in a climatology. The rank correlations of both resolutions remain below 0.5 given the difficulty of exactly represent the same 500 events in a 44-year climatology representing 3% of all considered days. Figure 6 also show relevant periods of heavy precipitation clustering, e.g., spring-summer of 1971, winter 1989, the years 2000 to 2002 and autumn 2013. Regarding EOBS-25km (grey squares), it shows a rank correlation of 0.94 against HYRAS-5m showing the good accuracy of





this product. Finally, the detection of cases in winter and summer in all datasets indicates that the PSI is a suitable method for extremes detection in all seasons.

## 5 Event scale evaluation

In the previous section, we assessed an overestimation of grid-point heavy precipitation for the convection-permitting simulation CPM, but a good performance in detecting severe precipitation events in a 44-year climatology. Here we evaluate the performance of CPM at the event scale. We focus on the period 2000-2015 to warrant the consistency between the analysed simulations al forced with ERA-interim (Tab. 2). The investigation region is ALP-3 (Fig. 1). Furthermore, the availability of radiosonde observations for validation is larger after the year 2000 in the UWYO dataset in ALP-3, which further supports our selection of the investigation period.

Table 3 shows eight subjectively selected events from the PSI extremes detection, also included in the derivation of the synoptic weather types in Sect. 3. Table 3 includes information about the duration of the events, the observed total precipitation, maximum grid point intensities, percentage of affected area, severity (PSI), and associated Weather Types (WT).

## 5.1 Precipitation

We evaluate the model performance focusing on two aspects of heavy precipitation, (1) the amount, calculated as aggregated precipitation in time and space, and (2) the structure, validated by means of the FSS metric (Sect. 2.3.3). For both metrics, we use MSWEP-11km (Tab. 1) as the observational reference, after coarse-graining all compared datasets to a common grid of 25 km. MSWEP-11km is used provided its large accuracy due to the inclusion of Rain Gauges (Beck et al., 2017) and since precipitation occurs to a large extent over the Mediterranean Sea, where HYRAS-5km and EOBS-25km have no coverage.

Table 4 shows the relative differences in precipitation amount aggregated in space and time between the model and observations as $RR_{rel.diff} = (MOD - OBS)/OBS$ in percent. CPM performed better than RCM in six out of the eight selected cases for precipitation amount. The largest improvement occurred for the 31-May-2013 event, which corresponds to the synoptic pattern S1 associated with the occurrence of ECOs and the advection of southwesterly moisture (Fig. 4) Using CPM brought generally larger precipitation rates, in agreement with the findings of Sect. 3, allowing for better scores of aggregated precipitation.

Regarding FSS CPM performed well, in general terms, for 7 out of 8 events with FSS reaching values over 0.7. RCM, for its part, performed well for 5 out of 8 events (Tab. 4). The 31-May-2013 event is again an example of good performance by CPM, where the FSS scores reached 0.87 (0.26 in RCM). The main reason for this improvement was the ability of CPM to represent larger precipitation structures over the Alps in a better agreement with MSWEP-11km. The spatial distributions of precipitation are shown in Fig. S1 (supplementary material).



Only the event 08-Aug-2007, showed a bad performance by CPM, both for precipitation amount and structure. This event
occurred under a S1 synoptic situation associated with an elongated troughs or cut-off lows (Fig. 4). The reason behind the
bad performance in this case in CPM is the larg underestimation of precipitation, which also hampers the structure
representation.

Overall, these results showed that CPM brings added value in the representation of precipitation amount and structure
compared to RCM. The advantage of CPM relies on the better location of orographic precipitation and the increased intensities
brought by the more intense updrafts and larger number of cells triggered.

## 5.2 Humidity and temperature

In addition to precipitation errors, temperature and humidity biases could affect our interpretation of the scale-dependency.
Here we validate specific humidity (*hus*) and temperature (*ta*) profiles from COSMO-CLM against radiosondes from the
University of Wyoming (UWYO) and surface relative humidity (*hurs*) against EOBS-25km for the eight selected events (cf.
Tab. 3).

Figure 7 shows the temporal Mean Bias (MB; thick line), the spread (shaded area), and the Root Mean Square Errors (RMSE;
dashed line) of specific humidity (Fig. 7a) and temperature (Fig.7b). The model output is interpolated to the location of eleven
sounding stations Only stations with a height difference lower than 50 m between the station height and RCM and CPM's
model orography are selected. This requirement is introduced to avoid including large humidity and temperature biases from
differences in surface topography between the model and the observations. We include all available soundings during the
duration of the eight events (Tab. 3) in the calculation, with a temporal resolution between 6 h and 12 h.

Humidity is slightly overestimated by RCM throughout the whole profile and by CPM above 800 hPa (Fig. 7a). The
overestimation by both models reaches 0.2 g kg$^{-1}$ at 700 hPa. Below 800 hPa, CPM, reduces the mean bias reaching -0.1 g kg$^{-1}$, indicating a generally drier planetary boundary layer. RMSE values are very similar for both simulations being close to 1.5
g kg$^{-1}$ below 700 hPa. These results highlight the difficulties of the COSMO-CLM model in representing an accurate
atmospheric humidity vertical gradient (Caldas-Alvarez and Khodayar, 2020; Caldas-Alvarez et al., 2021), which have been
observed in other, similar, non-hydrostatic models (Risanto et al., 2019; Bastin et al., 2019).

Regarding temperature (Fig. 7b), COSMO-CLM shows a warm bias, with mean bias reaching 0.5°C above 925 hPa for both
resolutions. RMSE (Fig. 7b, dashed line) is very similar between both simulations, close to 2 °C, with a slight improvement
by CPM (red). The temporal averaged profiles of specific humidity and temperatures for these data sets can be found in Fig.
S2 (supplementary material).

Provided the observations available below 925 hPa in the UWYO soundings were scarce, we employ the gridded EOBS-25km
dataset (Tab. 1) to investigate the COSMO-CLM biases at the surface (Fig. 8). We represent the spatial distribution of temporal
mean bias (colour shading) and the temporally-spatially averaged mean bias and RMSE of daily surface relative humidity. We





calculate relative humidity biases for this validation, given no surface specific humidity gridded observations with sufficient accuracy were available for our region and period of investigation.

COSMO-CLM underestimates surface relative humidity for both RCM (Fig. 8a) and CPM (Fig. 8b). This is especially so at the Po Valley (Italy) and the southern Italian Peninsula. However, CPM (Fig. 8b), slightly improves the surface relative humidity deficit at locations north of the Alps, e.g., northwestern France, the Czech Republic and western Austria. These

corrections in the northwestern part of the simulation domain, reduce the temporal and spatial MB by 3%. However, provided the larger spatial variability of this variable in CPM, due to the better orography representation, the RMSE is worsened by 5 %.

The profile and surface humidity and temperature validation has shown that: a) COSMO-CLM shows a bias for the humidity and temperature gradient with height, simulating a drier-than-observations surface level and a wetter-than-observed

atmosphere over 800 hPa both for RCM and CPM; b) COSMO-CLM presents a continuous warm bias, above 925 hPa; c) CPM reduces the positive surface relative humidity  bias over locations north of the Alps, e.g., western France, the Czech Republic and eastern Austria.

## 6 Scale dependency of thermodynamic processes

We analyse the scale-dependency of model variables related to thermodynamical processes of heavy precipitation in the

investigation area ALP-3. To this end, we use PCA as described in Sect. 2.3.2 with daily precipitation in the period 2000-2015 from RCM and CPM. We obtain the precipitation EOFs separately for winter (SONDJF) and summer (MAMJJA) for both resolutions. We derive composites of model variables related to thermodynamic precipitation processes, e.g. moisture transport, instability, or soil-atmosphere fluxes and we select precipitation days of each EOF with PC values larger than one standard deviation of the time series. This implies using data of about 400 days out of a time series of 2900 days associated to

each EOFs. Then the timely averages for each variable and EOF from RCM are subtracted to the temporal averages of CPM for the days prior to heavy precipitation days to derive composites of the resolution differences. Following Ulbrich et al., (1999), we use composites of timely averages, to avoid assuming linearity between the precipitation PCs and the variables' temporal evolution, as is the case when the Pearson correlation is calculated.

To study the scale-dependency of thermodynamic variables, we focus exclusively on precipitation EOFs with a similar

structure between RCM and CPM. This is done to ensure comparability between both resolutions. Being precipitation a highly variable quantity, winter and summer EOFs start to differ considerably after the fourth EOF for winter and the third EOF for summer. We do not consider the subsequent principal components. The leading four EOFs for winter explain 46% of the variability for the RCM and 42% for the CPM simulation. The first three EOFs for summer, for its part, explaining 39% of the situations in RCM and 33% in CPM). Here EOF-1 is presented for illustration (Fig. 9) but the remainder analysed EOFs can

be found in the supplementary material (Figs. S3, S4, and S5).





The first precipitation EOF has a very similar pattern between RCM and CPM (Fig. 9). For the winter season, both data sets (Figs. 9.a and 9.c) are dominated by orographic precipitation, over the Pyrenees (France), Corsica and the Central Massive (France), the Alps (Germany), and the Apennines (Italy). The differences of the composites (Fig. 9.e), clearly show larger precipitation in CPM (red) than in RCM (blue) over the mountain systems, as expected from the intensification of vertical

winds and more frequent triggering of convective cells (Langhans et al., 2012; Barthlott and Hoose, 2015). For lower terrain, CPM (red) only shows more precipitation north of the Alps than RCM. The opposite occurs for RCM (blue), showing larger precipitation over the Mediterranean Sea and the Po Valley (Italy). Spatially averaged, the CPM composites showed -0.14 mm h$^{-1}$ less than RCM (Fig. 9e). Chan et al., (2012) pointed at these differences between low and high terrain, arguing a lower skill of convection permitting models for lowlands.

For summer (Figs. 9b, 9d, 9f), a similar scale-dependency is found, with orographic precipitation dominating the total amount in RCM and CPM. The composite differences (Fig. 9f) show larger precipitation by CPM over the mountain tops. At low terrain, the summer composites show larger precipitation by RCM towards the northern Alpine region (Austria, southern Germany; Fig. 9b). The spatial average of precipitation differences for this season is -0.12 mm h$^{-1}$, again with CPM showing larger precipitation than RCM.

The findings based on the main modes of precipitation variance, for which EOF-1 is shown as an example, can be summarized as follows: (a) CPM displays larger precipitation than RCM over the mountains for all assessed EOFs and winter and summer seasons. This points at resolution differences in dynamic processes, e.g., increased vertical wind speeds, larger triggering of convective cells (Langhans et al., 2012; Barthlott and Hoose, 2015); to be the main precursors of the differences; especially since EOF-1 is the most frequent precipitation mode. (b) Over lowlands and the Mediterranean, the dynamical factor alone

cannot explain the precipitation differences.

To understand precipitation differences at low terrain due to model resolution, Fig. 10 shows composite differences for model variables related to thermodynamic precipitation processes, relative to EOF-1. For the calculation of the composite differences, we use the day prior to the selected precipitation days to study the preconditioning of the precipitation environment.

Figure 10.a shows winter differences in specific humidity at the surface (*huss*; colour shading), and precipitable water vapour

(*prw*; contours). The surface specific humidity differences show a north-south gradient, changing from higher surface humidity in CPM north of the Alps (red), up to 0.4 mm, to higher humidity in RCM (blue) over the Alps, Italy, and the Mediterranean Sea (0.8 mm). Regarding precipitable water vapour, RCM (positive contours) represents systematically a wetter atmosphere, especially over Italy and the Mediterranean Sea with differences up to 1 mm in compared to CPM. This holds for all four analysed winter EOFs. Figure 10.c shows composite differences of CAPE (colour shading) and Equivalent Potential

Temperature at 850 hPa (isolines). CAPE is larger over the Mediterranean for RCM (blue) up to 80 J Kg$^{-1}$ more, but larger for CPM (red) over western France. $\theta_e^{850}$ is 1 K larger in RCM over the Adriatic Sea and eastern Italy. This is probably due to the larger moisture amount in RCM at the lower mid-troposphere. Fig. 10.e, shows differences in outbound sensible heat flux





($hfss$; colours) where green colours denote grid points in RCM or CPM with no outbound fluxes of sensible heat. The contours in Fig. 10c represent surface temperature at 2 m ($tas$). Due to the colder soil conditions in winter, RCM and CPM show

predominantly a surface-directed flux of sensible heat over land (green colours). Over the Sea, the atmosphere-directed surface flux of sensible heat is dominated by RCM with an excess up to 10 W m$^{-2}$. Surface temperature is larger by 1 K in RCM compared to CPM over land with no differences over the Sea. Finally, Fig. 10.g shows differences in outbound latent heat flux ($hfls$, colour shading) and wind speeds at 10 m height ($sfcwind$; contours). Differences in atmosphere-directed latent heat fluxes show a marked land-sea contrast with larger emissions over land for CPM (red) but larger emissions over the Sea for

RCM (blue). This agrees well with the differences observed in outward surface sensible heat flux which show an opposite sign to keep a similar Bowen's ratio over land. RCM shows about 0.2 m s$^{-1}$ higher 10m wind speeds at the surface over the Sea and the Italian peninsula, whereas CPM shows higher winds over the Alps. In the northern part of the domain, the differences in the wind speed are negligible.

Figure 10.b shows the results for summer EOF-1 composite differences. Analogously to winter, RCM (blue) shows larger

surface specific humidity, up to 1 g kg$^{-1}$ more than CPM (red) which affect the whole investigation domain except northern Europe (Fig. 10.b). Precipitable water vapour is analogously larger in RCM than in CPM with differences up to 1.5 mm in the complete domain. These differences are larger in summer composites than in winter because of the larger water capacity of warmer air masses. Figure 10.d shows larger CAPE by RCM (blue) along the Adriatic, the western Mediterranean and the Po Valley (Italy), whereas CPM (red) represents larger CAPE north of the Alps and the Balkans with no differences over the

Apennines (Italy). On the contrary, differences in $\theta_e^{850}$ (Fig. 10.d, contours), show an excess in RCM (positive contours) over the whole domain, up to 3 K. Figure 10.f shows that RCM represents larger outbound surface sensible heat fluxes over all land areas up to 30 Wm$^{-2}$ (blue colours), with no remarkable differences over the Sea, similarly to winter EOF-1. For surface temperature (contours), the warmer surface level in RCM due to enhanced emission of sensible heat occurs likewise in summer, although with larger differences (1.5 K over western France). Finally, Fig. 10.h, shows larger outbound latent heat fluxes

(colour shading) in RCM (blue) over the Sea but larger emissions of latent heat in CPM (red) over land, with the only exception of the Po Valley. Wind speed at the surface (contours in Fig. 10.h) is larger in RCM than in CPM over northern France, but the opposite occurs, over the Alps and the northwestern Part of the Mediterranean (south to France). Over the Apennines, the differences are negligible.

The analysis of the composites showed new insights on how precipitation differences relate to the scale-dependency of

thermodynamic processes. We presented the results of EOF-1 for illustration (Fig. 10) but the remainder EOFs are included in the supplementary material (Figs. S6, S7, and S8). Our findings indicate for all EOFs in winter and summer that: a) precipitation differences due to resolution at lowlands are related to the differences of the thermodynamic variables analysed here b) the variable whose scale-dependency showed the largest impact on precipitation differences at lowlands was surface specific humidity. Larger surface humidity in RCM or CPM in the day prior to the event, determined whether RCM or CPM

represented larger precipitation; c) precipitation differences for EOF-3 and EOF-4 in Winter and EOF-1 in Summer could not



be explained through resolution differences in surface specific humidity For those EOFs, RCM represented larger precipitation, if the events were preconditioned by very large $\theta_e^{850}$ differences (3 K or more) as is the case for EOF-4 in winter and EOF-1 in summer, whereas CPM simulated larger precipitation where the preconditions showed larger CAPE and stronger surface winds in CPM as in EOF-3 in winter; d) generally, CPM represented a wetter surface north of the Alps and RCM represented

a wetter surface over the Mediterranean Sea and the Italian Peninsula; e) the surface specific humidity differences can be explained through differences latent heat fluxes between RCM and CPM, where RCM evaporates more moisture over the Sea and CPM over land. Provided the predominant southwesterly to southeasterly flow in the Mediterranean region transporting moist air masses (Toreti et al., 2010), the Italian peninsula predominantly showed a wetter low level in RCM compared to CPM; f) outbound sensible heat fluxes and surface temperature over land, were systematically larger in RCM compared to

CPM; f) CAPE was systematically larger in CPM over land and surface winds were stronger in CPM over the Sea and the Alps.

## 7 Conclusions

In this study we evaluated reanalysis-driven simulations of the greater Alpine area in an RCM (parameterized deep convection at 25 km) and a CPM (explicitly resolved deep convection at 3 km) setups, to assess the scale-dependency of thermodynamical

processes influencing extreme precipitation. The main results are:

a) The implemented Precipitation Severity Index (PSI) considering heavy grid point precipitation intensities, with large spatial coverage and persistence is a suitable index for extremes detection both for summer and winter periods.

b) Winter heavy precipitation events in the 1971-2015 period in the greater Alpine area occur either under stationary front situations with northerly low pressure descending to central and Southern Europe (EOF 1) or under strong north-south

gradients of the 500 hPa geopotential height with a strong zonal flow (EOFs 2, 3, and 4). Four principal weather types suffice to explain the major part of the natural variability of winter cases.

c) Summer events are associated to either frontal convection on the western sector of elongated upper-level troughs and evolved cut-off lows (EOFs 1, 3 and 4), or due to winter-like synoptic patterns of stationary fronts over central Europe or strong zonal flows (EOFs 2 and 5). Five PCs are sufficient to explain the major part of the natural variability of summer cases.

d) The CPM set up shows larger precipitation intensities, better rank correlation, better hit rates for extremes detection, and a better representation of precipitation amount and structure for the selected HPEs, compared to RCM. However, CPM overestimates grid point intensity, especially at high altitudes such as the Alps or the Apennines (also observed in Langhans et al., 2012, and Ban et al., 2020). However the observations over mountainous terrain might as well be strongly underrepresented due to a lack of measurements to account for the high spatial variability..





e) The scale-dependent precipitation differences over elevated terrain are explained by differences in the dynamical factors of deep convection, i.e., intensification of the updraughts, and larger triggering of convective cells in CPMs assessed in previous studies (Langhans et al., 2012; Vanden Broucke et al., 2018; Heim et al.2020).

f) Conversely, precipitation differences over low terrain are mostly related to the scale-dependency of thermodynamic variables. Among the analysed variables, surface specific humidity, showed the largest impact. An excess for this variable up 500 to 1 g kg $^{-1}$ on the day prior to precipitation in RCM (CPM) induced larger precipitation totals at low altitudes in RCM (CPM). Since RCM showed generally larger surface humidity over the Mediterranean and the Italian peninsula, and CPM showed larger surface moisture north of the Alps, precipitation differences at low elevations showed generally a north-to-south spatial distribution.

g) The larger surface specific humidity in RCM over the Mediterranean and the Italian Peninsula, and the larger surface 505 humidity in CPM north of the Alps are due to resolution differences in outbound latent fluxes from the soil to the atmosphere. RCM showed an intensified emission of latent heat over the Mediterranean Sea (25 W m$^{-2}$), whereas CPM, showed larger latent heat emission over land (15 W m$^{-2}$). The southerly winds, over the Mediterranean and Adriatic Seas induced an inland transport of surface atmospheric moisture through the Po valley and the Italian peninsula. These resolution differences in latent heat fluxes are responsible for the slight improvement in the simulation of surface relative humidity assessed for CPM (3 % 510 less mean bias), north of the Alps.

h) Contrary to latent heat, RCM emits more sensible heat fluxes over land than CPM, up to 40 W m$^{-2}$. This induces larger surface temperatures in RCM up to 1.5 °C compared to CPM. The differences over the Sea are negligible.

Our study presents new insights on the scale-dependency of precipitation thermodynamic processes. However, new research questions remain open that should be subject to further investigation. The underestimation of moisture at the surface at both 515 resolutions (RCM and CPM), which is slightly improved north of the Alps in CPM must be assessed further. More knowledge is needed as to whether other established regional climate models present similar shortcomings. Furthermore, this work points at the scale-dependency of surface humidity as a precursor of precipitation differences over low terrain. However, more research is required as this general response to surface humidity differences could not be asserted for all analysed precipitation modes, namely EOF-3 and EOF-4 in Winter and EOF-1 in Summer. For these EOFs, the scale dependency of CAPE, $\theta_e^{850}$ 520 and surface wind speed played a larger role. In-depth analysis for individual HPEs is encouraged to assess these model responses. Finally, our work highlights the existent dichotomy between low vs. high terrain and dynamic vs. thermodynamic factors of the scale-dependency representation of extreme precipitation. This aspect should be considered in future scale-dependency studies of the greater Alpine area and other orographically complex areas.

Our study has some limitations that need to be briefly addressed. First, we only assess one regional climate model and hence 525 our results cannot be generalized to other RCMs. Besides, we performed precipitation and humidity validations only for eight selected case studies, an insufficient number to derive statistically robust conclusions for the climatology. However, this



evaluation provide a first estimation of the overall model's performance. Finally, the combination of two different simulations (KLIWA-2.8 km and ALP-3 from the FPS-Convection project) to build the CPM are not identical (Tab.2).We decided nevertheless to combine both simulations to profit from a larger time span, that enables studies on climatic scales.

Notwithstanding these limitations, our study provides evidence of the added value of using CPM for climate studies. Despite the assessed positive model bias for mountain heavy precipitation, the good results in the event validation and detection capabilities in the climatology, as well as the improved description of the physics, emphasize their applicability in both climate and impact studies. Examples of endeavours where high-resolution climate data bring added value are, for instance, the downscaling of climate change projections (Pichelli et al., 2021), the development of decision-relevant strategies for Climate

Change adaptation (BMBF-RegiKlim) or their use in forestry or hydrology applications interesting for the scientific community and stakeholders.



## 8 Figures and tables

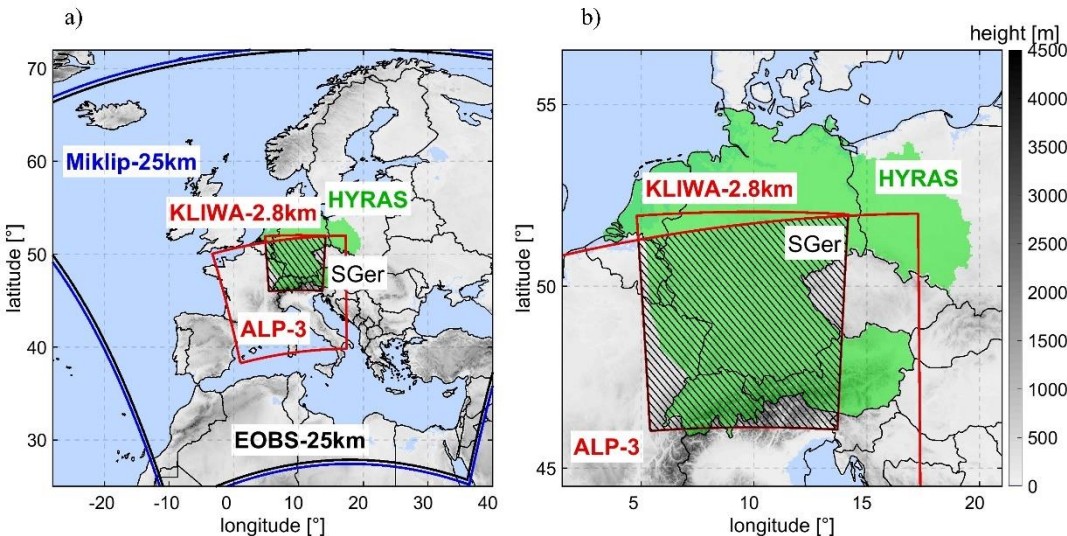


**Figure 1. a) Simulation domains of Miklip-25km (blue), KLIWA-2.8km (red), ALP-3 (red), and observations' coverage of HYRAS-5km (green), and EOBS-25km (black). The two investigation domains of this study are Southern Germany (SGer; dashed box), and the ALP-3 domain.**

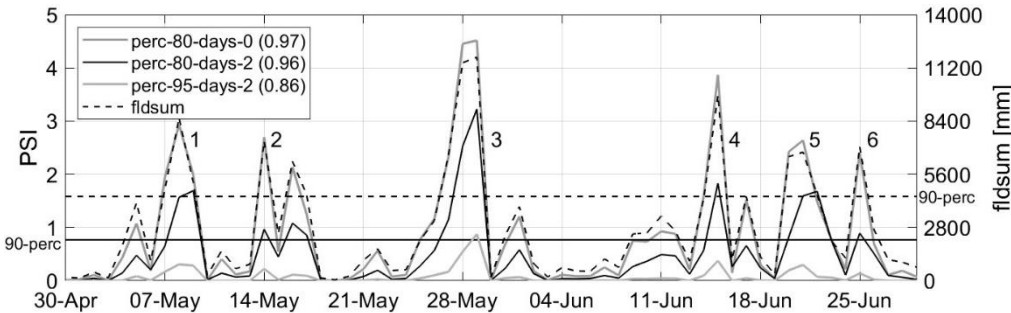


**Figure 2. Daily PSI (left axis) and field sums (fldsum; right axis) over the SGer investigation area between 30 April and 29 June 2007 based on HYRAS-5km. Different settings of the percentile ($RR_{perc_{ij}}$) and persistence ($d$) parameters in the PSI calculation (Eq. 1) are tested. Namely, the 80th percentile and a persistence of 0 days (perc-80-days-0; dark grey), an 80th percentile and a persistence of 2 days (perc-80-days-2; black), and a 95th percentile and a persistence of 2 days (perc-95-days-2; light grey). For comparison,**
**fldsum is shown (dashed black line). The legend shows in brackets the Spearman's rank correlation of the different PSI calculations with fldsum. The 90th percentile values of the PSI (left) and fldsum (right) for the whole 1971-2015 climatology are shown by horizontal lines. Six HPEs denoted by numbers take place during the shown period, the 7-11 May event over northern Germany and the Alps (1), the 14-17 event affecting northern Poland and the Alps (2), the 27-29 May convective events over central Germany and the northern Alps (3), the 15 Jun event over central Germany (4), the 20-21 June over northern Germany and eastern Poland (5),**
**and the 25-Jun Alpine event (6), see Schacher et al., (2007).**



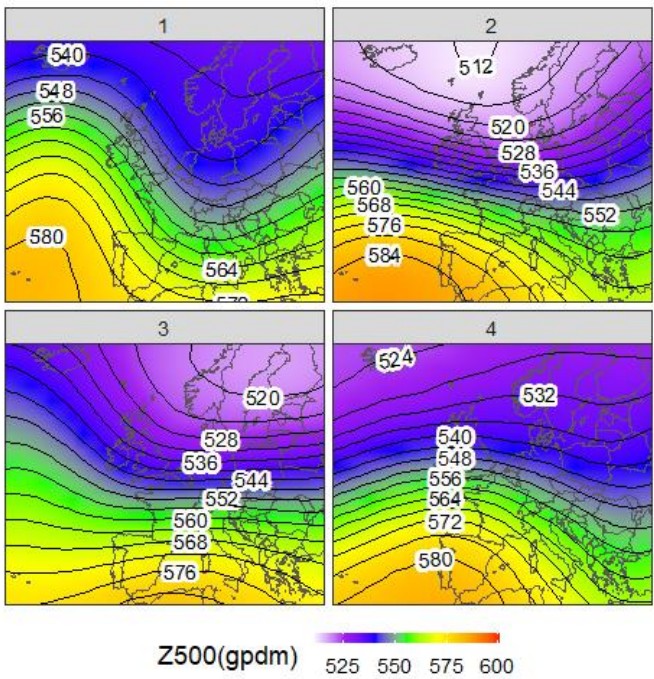

**Figure 3. Synoptic classification based on Principal Component Analyses for the 98-percentile most severe precipitation cases in winter (SONDJF) of the 1971-2015 period, detected with the PSI. The selection of the number of PCs is based on parallel analyses and randomization of the distribution eigenvalues ($\lambda$; Peres-Neto et al., 2005). The spatial distributions show 500 hPa geopotential height in geopotential decameters (gpdm) obtained from RCM. The analysis has been performed with the SynoptReg R package (M. Lemus-Canovas et al., 2019).**






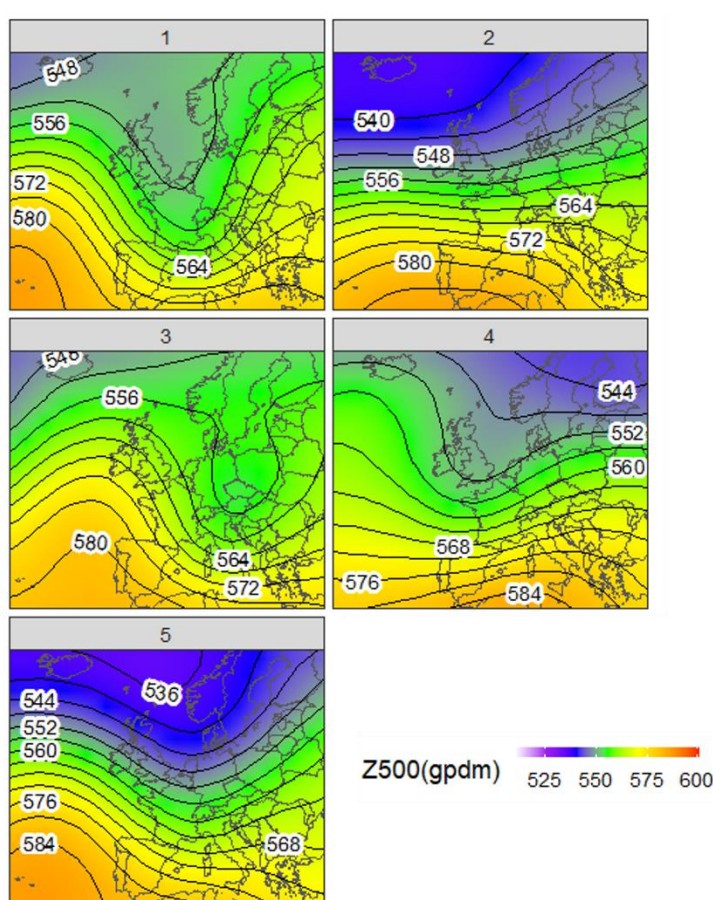

**Figure 4. As Fig. 3 for summer extreme precipitation days (MAMJJA). 5 PCs are discernible from random noise.**





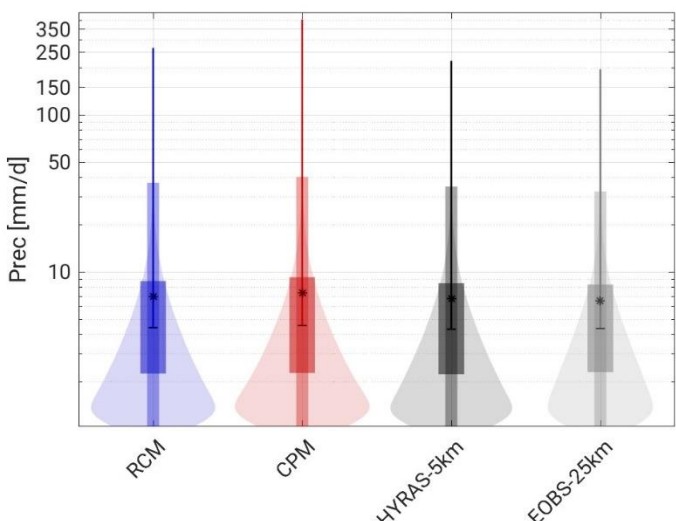

**Figure 5.** Box and whisker plots showing daily precipitation median (horizontal bar), mean (asterisk), upper and lower quartiles (boxes), 1- and 99-percentile (vertical bars), and maximum grid point precipitation (vertical line) in the period 1971-2015 over SGer. The kernel density at each precipitation intensity is shown by the shaded areas. All data sets have been previously upscaled to a common grid of 25 km by means of conservative remapping.





**Figure 6. Dot diagram of the period 1971-2015, showing the 500 most severe precipitation events, detected using the PSI. The results are shown for HYRAS-5km (black circles), EOBS-25km (grey squares), RCM (blue circles), and CPM (red dots). The spearman's rank correlation of the data sets is shown in the legend whereHYRAS-5km taken as the reference. All daily precipitation datasets have been upscaled to a common grid of 25 km by means of a conservative remapping**

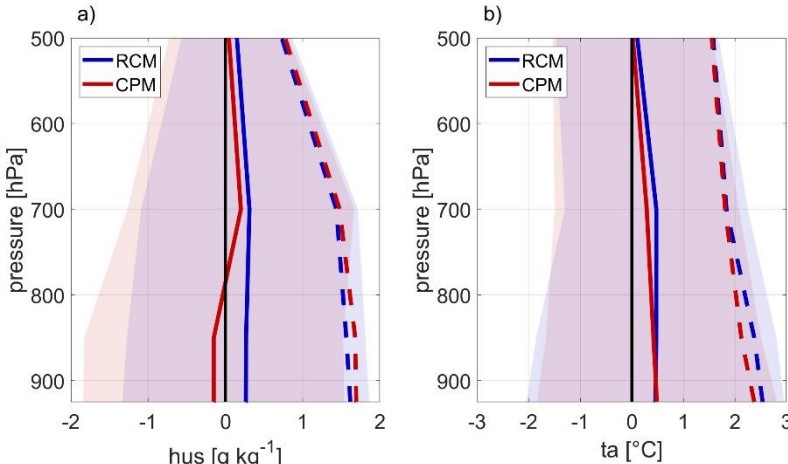

**Figure 7. Validation of the model's specific humidity (a) and temperature (b) against radiosondes obtained from the UWYO soundings All available soundings during the 8 selected events (Tab. 3) are used for the validation. In total, 11 stations within the ALP-3 simulation domain (Fig. 1) are used, namely Nimes (France); Oppin, Meiningen, Idar-Oberstein, Stuttgart, Kümmersbruck**
**and Munich (Germany); Praha (Czech Republic); Milano, S. Pietro, and Pratica di Mare (Italy). The model information is interpolated to the station location. The MB is calculated as MOD-OBS, hence positive MB values, indicate an overestimation of humidity or temperature.**

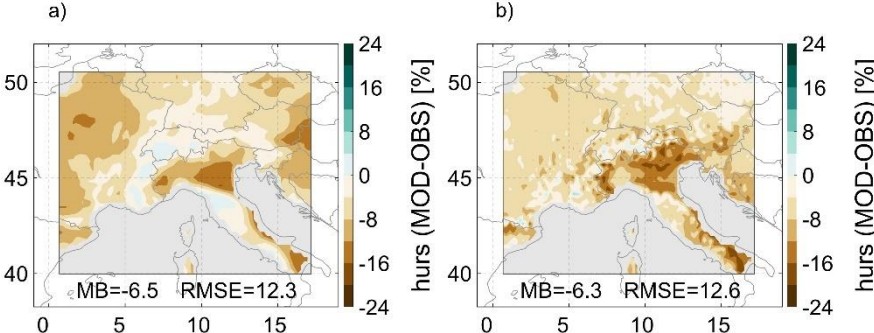

**Figure 8. Spatial distributions of hurs Mean Bias (MB), obtained as differences between (a) RCM and EOBS-25km and (b) between CPM and EOBS-25km. All datasets have been coarse-grained to a 25 km resolution common grid. The spatially averaged MB and Root Mean Squared Error (RMSE) is shown in text. MB and RMSE are obtained from daily hurs values for all days in the 8 selected events (Tab.3).**

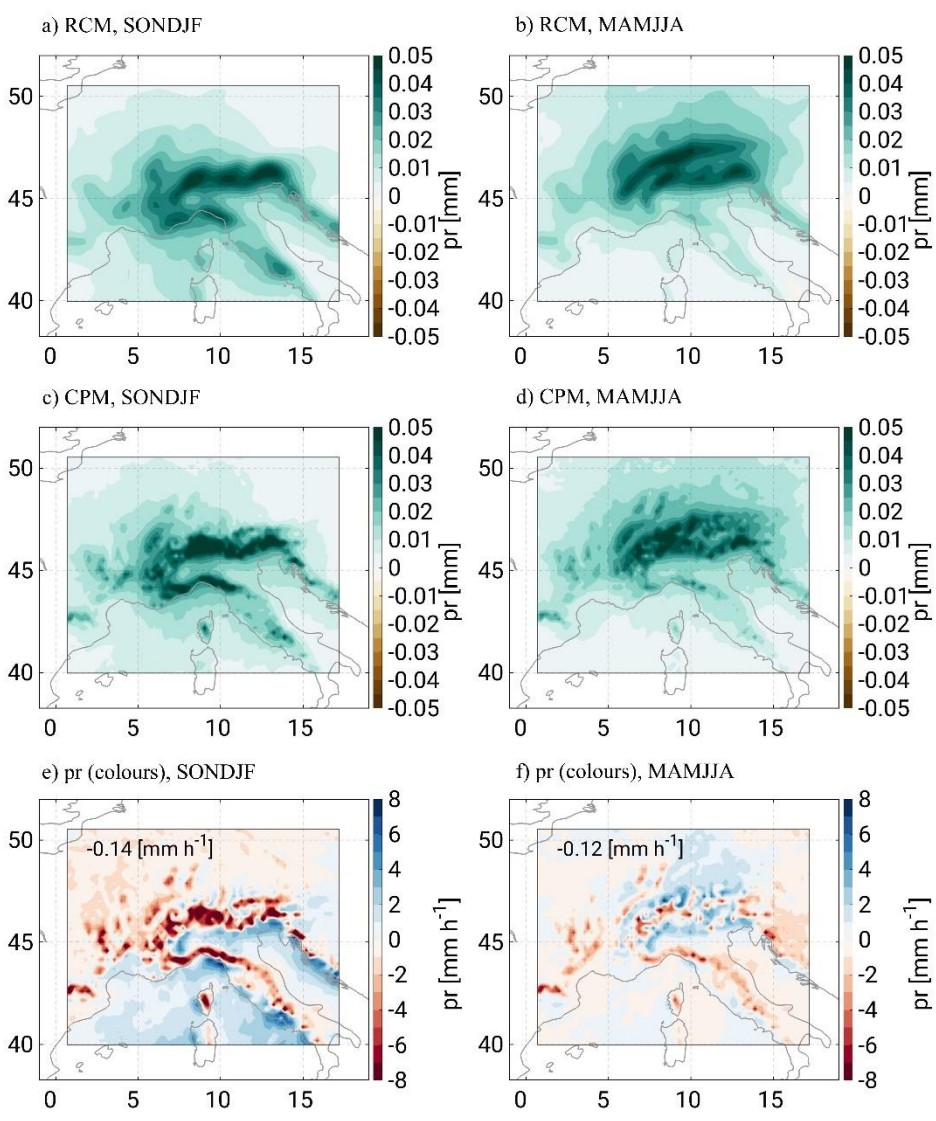


**Figure 9. Empirical Orthogonal Function 1 of precipitation for SONDJF (a, c, e) and MAMJJA (b, d, f) for the RCM (a, b) and CPM (c, d) simulations. The EOF-1 is obtained using daily precipitation values in each season (SONDJF and MAMJJA) in the period 2000-2015. The composite differences (e, f) are calculated as differences of the timely averages of precipitation on days showing PC values larger than one standard deviation, i.e, days showing a large similarity to the spatial distribution of EOF-1.**


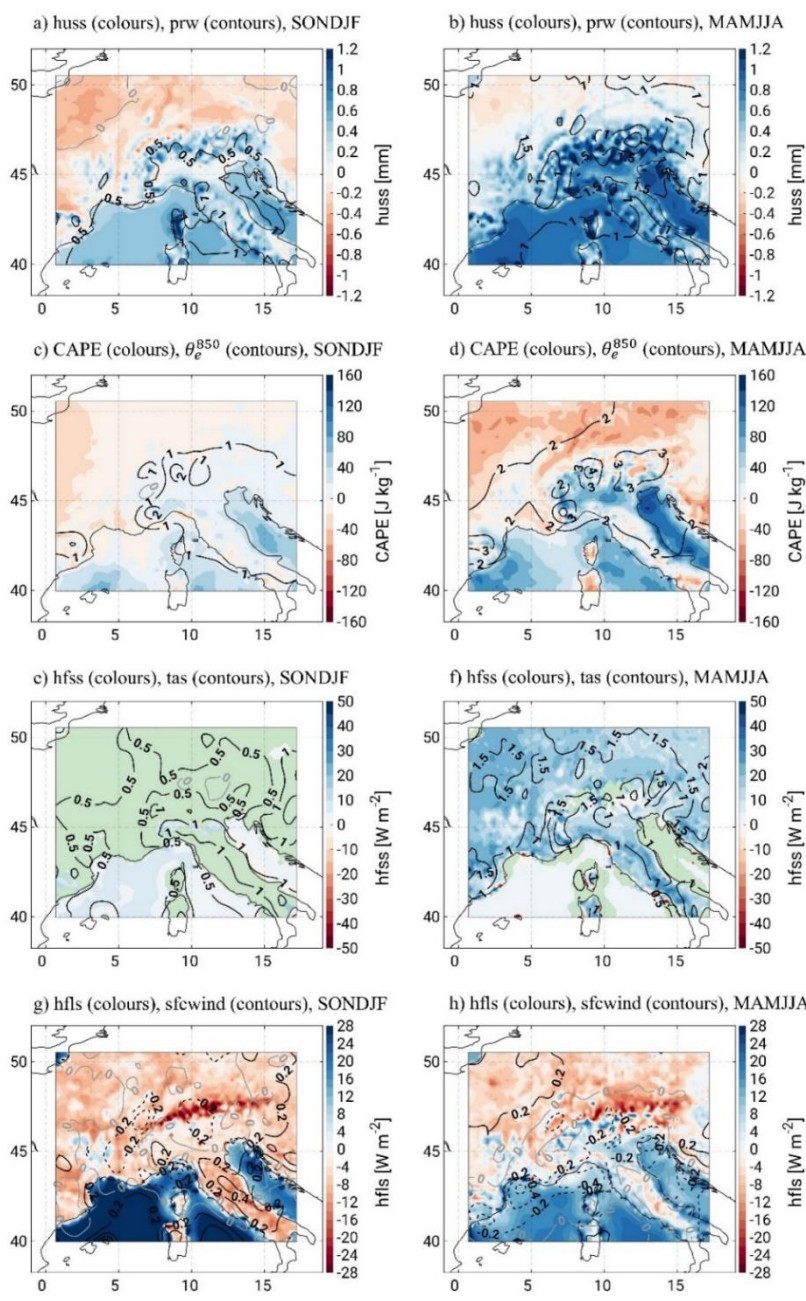

**Figure 10.** Composite differences of thermodynamical variables associated with heavy precipitation on the day prior to selected precipitation dates. The precipitation days are selected as those over one standard deviation of the Principal Components for EOF-1 shown in Fig. 9. Positive contours or blue colours stand for larger amount in RCM, negative contours and red colours for larger amount in CPM. The variables represented are (a, b) surface specific humidity ($huss$), precipitable water vapour ($prw$); (c, d); Convective Available Potential Energy ($CAPE$), equivalent potential temperature at 850 hPa ($\theta_e^{850}$); (e, f) outbound surface sensible heat flux ($hfss$) and surface temperature ($tas$); (g, h) outbound surface latent heat flux ($hfls$) and surface wind speed ($sfcwind$). Green colours in $hfss$, denote grid points with no outbound sensible heat flux in RCM or CPM.



**Table 1. Description of observational data sets used for validation of the simulations. The observational data set types used to create the different products are Radar (R), Gauges (G), Satellites (S), and Reanalysis (R).**

| Name | Vers. | Res. | Per. | Observations | Provider | Reference | Cover. |
|---|---|---|---|---|---|---|---|
| EOBS-25km | v20.0e | 25 km, daily | 1950-2020 | Rain Gauges (G), surf. rel. humidity (*hurs*) | ECAD | Cornes et al., (2018) | Europe |
| HYRAS-5km | v2 | 5 km, daily | 1951-2015 | Rain Gauges (G) | DWD & BfG | Rauthe et al., (2013), Razafimaharo et al. (2020) | Germany |
| MSWEP-11km | v2.2.0 | 11 km, 3-hly | 1979-2020 | CPC (G), GPCC (G), CMORPH (S), TMPA-3B42RT (S), GSMaP (S), ERA-Interim (R), JRA-55 (R) | GloH2O | Beck et al., (2017) | Global |
| UWYO | - | stations, 12 hly | 2000-2015 | Radiosondes | Wyoming Univers. | http://weather.uwyo.edu/upperair/sounding.html | Global |

**Table 2. Reanalysis-driven COSMO-CLM decadal simulations.**

| Name | Res. | Convect. | Lev. | Forcing | Period | Project | Domain |
|---|---|---|---|---|---|---|---|
| RCM | 25 km, 3-hly | Tiedke Deep + Shall. | 40 | ERA-40 | 1961-1979 | Miklip-II | Miklip-25km |
| | | | | ERA-int | 1980-2018 | | |
| CPM | 2.8 km, 1-hly | Tiedtke Shall. | 49 | ERA-40 | 1971-1999 | KLIWA[1] | KLIWA-2.8km[2] |
| | 3 km, 1-hly | | 50 | ERA-int | 2000-2015 | FPS-Convection | ALP-3[3] |

[1] Simulations provided by the KLIWA project (www.kliwa.de: Hackenbruch et al., 2016)
[2] Domain covers southern Germany, Switzerland, and the eastern Czech Republic.
[3] Domain covers France, northern Italy, Switzerland, the Czech Republic, and southern Germany.

**Table 3. Selected eight heavy precipitation by means of the PSI between 2000-2015. The area considered is area SGer (see Fig. 1). The synoptic Weather Types (WT) have been obtained using PSML data from RCM (Sect. 3). The PSI values, total precipitation, maximum grid point precipitation and coverage are obtained from HYRAS-5km.**

| Event | Event days | Total. Precip. [mm] | Max. prec. [mmd⁻¹] | Coverage [%] | PSI | WT |
|---|---|---|---|---|---|---|
| 15-Jul-2001 | 12-16 Jul | 81098 | 141 | 83 | 2.22 | S2 |
| 03-Nov-2002 | 2-5 Nov | 80592 | 52 | 96 | 2.55 | W4 |
| 13-Jan-2004 | 11-15 Jan | 97706 | 103 | 97 | 3.62 | W4 |
| 22-Aug-2005 | 19-23 Aug | 106852 | 177 | 80 | 2.31 | S4 |
| 08-Aug-2007 | 07-09 Aug | 85473 | 95 | 89 | 2.79 | S1 |
| 31-May-2013 | 31 May-02 Jun | 77958 | 99 | 94 | 3.24 | S1 |
| 08-Jul-2014 | 06-13 Jul | 155621 | 83 | 99 | 3.21 | S1 |
| 20-Nov-2015 | 19-21 Nov | 102747 | 109 | 82 | 2.83 | W1 |





**Table 4. Scores of precipitation validation.** $RR_{rel.diff.}$ **stands for the relative differences of spatially and temporally aggregated**
**precipitation between the model and observations for the duration of each event (see Tab. 3). The relative differences are calculated**
**as** $(RR_{mod} - RR_{obs})/RR_{obs}$ **and are given in percentage. The negative signs imply an underestimation of precipitation in the model.**
**FSS is the Fractions Skill Score between the model and the observations (Sect. 2.3.3), again MSWEP-11km is used as reference. The**
**temporal aggregation is 24 hours. The best scores are shown for FSS values closer to 1. All datasets are upscaled to a common grid**
**of 25 km. The investigation area is ALP-3 (Fig.1).**

| Event | $RR_{rel.diff.}$ [%] | | FSS | |
|---|---|---|---|---|
| | RCM | CPM | RCM | CPM |
| 15-Jul-2001 | -40 | -34 | 0.63 | 0.78 |
| 03-Nov-2002 | -16 | -11 | 0.81 | 0.82 |
| 13-Jan-2004 | -7 | -1 | 0.97 | 0.97 |
| 22-Aug-2005 | -28 | -26 | 0.88 | 0.83 |
| 08-Aug-2007 | -52 | -66 | 0.63 | 0.33 |
| 31-May-2013 | -44 | -5 | 0.26 | 0.87 |
| 08-Jul-2014 | -6 | -21 | 0.96 | 0.9 |
| 20-Nov-2015 | -18 | -17 | 0.92 | 0.93 |




## 9 Code availability

The COSMO-CLM is available for member of the CLM community and the documentation is accessible at, http://www.cosmo-model.org/content/model/documentation/core/default.htm (last accessed, 11-Aug-2021).

## 10 Data availability

The EOBS-25km dataset is accessible after registration at https://www.ecad.eu/download/ensembles/download.php#version (last accessed, 17-Dec-2021). The HYRAS-5km data set is publicly accessible at the Climate Data Centre (CDC) of the German Weather Service (DWD) at https://opendata.dwd.de/climate_environment/CDC (last accessed, 17-Dec-2021) MSWEP-11km, has been provided by the Climate Prediction Centre, after agreement of use. The soundings from UWYO are

publicly accessible at http://weather.uwyo.edu/upperair/sounding.html (last accessed, 17-Dec-2021). Further information about the XCES tool can be found in (https://www.xces.dkrz.de/)

## 11 Author contribution

ACA, HF, and JGP designed the study. ELE implemented the PSI index in the Mistral at the DKRZ. ACA and HF analysed the data. ACA prepared the figures and wrote the initial draft. All authors contributed with discussions and revisions.

## 12 Competing interests

The authors declare that they have no conflict of interest.

## 13 Acknowledgements

The research was accomplished within project A1 "SEVERE - Scale Dependent Process Representation and Sensitivity Analysis for Most Extreme Events" (Grant No. 01 LP 1901 A) and D2 "COSOX – Coordination of Software Management"

(Grant No. 01 LP 1904 B) within the German Federal Ministry of Education and Research (BMBF) research initiative ClimXtreme. JGP thanks the AXA research fund for support. The simulations were partly performed within the BMBF project MiKlip (FKZ: 01 LP 1518 A) at the German Climate Computing Center (DKRZ) and partly within the framework of CORDEX FPS Convection at the HLRS Stuttgart. We would like to thank Deborah Niermann and Stella Steidl at the (German Weather Service, DWD) for providing the HYRAS data set. We would like to thank Hylke Beck for sharing the MSWEP precipitation

data. Moreover, we would like to acknowledge M. Lemus-Canovas for providing the SynoptReg R package used for computing the synoptic weather types (Lemus-Canovas et al., 2019). We acknowledge the contribution of the DKRZ for storing and maintaining the model data and the FUB for the software coordination of XCES.





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
