# Peer review of "Convection-Parameterized and Convection-Permitting Modelling of heavy precipitation in decadal simulations of the greater Alpine region with COSMO-CLM"

_Weather and Climate Dynamics, 2022_

## Referee Comment (RC1)

**Anonymous review for the manuscript "Scale-dependency of extreme precipitation processes in regional climate simulations of the greater Alpine region" by Caldas-Alvarez et al. doi.org/10.5194/wcd-2022-11, 4.5.22**

The presented manuscript assesses differences in the statistics of daily ("heavy") precipitation between a 25km RCM simulation and two kilometer-resolution CPM simulations. This analysis has been done before, the results are quite robust across model codes and well described in the literature. The presented analysis is novel nevertheless, but resorts to rather sophisticated methods (PCA, composites) and indices (PSI, FSS, wet-day percentiles), making them rather complex and difficult to interpret. Unfortunately, some of their implementation details are not fully fit for purpose (see major issues below). Also, I am not convinced that the presented results allow supporting the interpretations and conclusions made at the end of the manuscript. In particular, I criticize some of the inferences made from the detected differences (see major issues below), albeit alternative hypothesis are discussed in the literature. Finally, the detected differences are rather small, while the necessary statistical quantification is not provided (E.g., L490 and L303). In my role as a reviewer, I usually abstain from requiring stat. tests, but when an established and robust hypothesis (no differences in daily statistics) is refuted in a manuscript, requiring a robustness assessment seems warranted.

I have the impression that the manuscript title does not reflect the content of the manuscript precisely enough, as neither "scale-dependecy" nor "extreme precipitation processes" are actually assessed in the study. In particular, I don't think that the chosen indices and events qualify as "extreme precipitation", and "extreme precipitation processes" are not considered at all. Meanwhile, the term "scale-dependecy" is usually used differently from the presented use case.

Overall, the simulation configurations are not described in sufficient detail and some important aspects are missing. E.g., for the ALPS-3 simulation, the authors mostly refer to Coppola et al., (2018), which is a MIP overview paper, and thus does provide the necessary detail to ensure scientific reproduceability. For instance, the info that KLIWA-2.8 and ALPS-3 were conducted using two different major releases of the code was not highlighted. I consider this detail is relevant context when suggesting that ALPS-3 is merely a continuation of KLIWA-2.8.

Please find below a list detailing the major and minor issues.

**Major Issues**

L153: "In spite of these small inconsistencies, we combine both CPM simulations to attain a sufficiently large investigation period for comparison with the RCM simulation and observational datasets."

I don't think that the term "small inconsistencies" is justified here. The KLIWA-2.8 and ALP-3 configurations differ in virtually every aspect, apart from their overlapping computational domains, and their use of the COSMO code (in two different major releases!).

Also, what defines "sufficiently large"? KLIWA-2.8 is 29 years long. I'd consider that sufficient for all of the presented analysis and the qualitative conclusions of the study. Note that KLIWA-2.8 is actually only used in Section 4 (Evaluation of extreme precipitation).

L170: Does the 80th all-days-percentile really characterize "high grid-point intensity"? I think the 80th percentile represents a value of a few mm/day, which is rather typical for a rainy day in Germany.

L193: I don't understand. In Fig. 2 the 11$^{th}$ Jun event is below the 90$^{th}$ percentile of both respective indices, no? This is exactly opposite to the argument made in the text.

L198: I don't think the analysis supports this claim, rather the opposite. As indicated in Fig. 2 the Spearman's rank correlation between PSI and fldsum is 0.98/0.96. Also it is evident that the solid line mostly tracks the dotted line. That is, at lower amplitude, which does not matter for a ranked index. This means that after applying the second threshold (i.e., L236, L301) almost the exact same events are chosen as would have been when using the fldsum. In other words, Fig. 2 actually demonstrates that PSI is an unnecessarily complex choice for the presented use-case.

L240: Why are the EOFs computed using the RCM and not ERA-Interim directly? My point is that the first few EOFs of the 500 hPa geopotential need to be almost exactly the same in RCM and driving data. After all, if those EOFs of the RCM simulation would become systematically different than those in the driving data, the RCM approach would become somewhat questionable.

L285: The analysis presented in Figure 5 is rather difficult to interpret, since the boxplot parameters (median, quartiles, ...) depict percentiles of a conditional index (i.e., the wet-day precip., P > 1 mm/day). Please consult the following study for a thorough discussion of the problems related to deriving percentiles of conditional indices: Schär et al., (2016)

L327: Why are events chosen that entail bad quality/non-existent observations (since over ocean)? To verify a model, I would intuitively choose case studies that have abundant high-quality observations available. Also, I am not fully convinced about the usefulness of the MSWEP-11km product.

L361: The statement needs to be qualified, also w.r.t. internal variability and accuracy of specific humidity obs with radiosondes. In fact, I was surprised to see only a difference of 0.1 – 0.2 g/kg when comparing profiles of a limited-area climate simulation to more-or-less "instantaneous" soundings. I think, my view is corroborated well when considering Fig. S2, instead of the differences. In contrast to the authors, I think these results are actually rather promising.

L384ff: I am not completely sure if the chosen procedure is appropriate, but honestly, I do not fully understand why it has been chosen in the first place. First of all, why would EOF1 be associated with "heavy precipitation"? I thought that EOF1 portrays the mode with the largest variance, right? That is (by definition) rather unlikely a percentile at the tail of the precip. Distribution, no? Second, EOFs seems rather complicated provided that the results (Fig. 9) exert a very similar pattern as much simpler indices, e.g., the standard deviation (cdo timstd).

L391: Why for the day prior? I thought the authors suggest the detected heavy precipitation events to be primarily associated with synoptic systems.

L398/399: If the first three leading EOFs only explain 39% of the variance. Is that analysis really an appropriate tool? Maybe I do not understand precisely what the authors are trying to achieve.

L415ff: How come? Why not a parameter choice, or a time-step sensitivity in the cloud microphysics parameterization, or the nesting strategy …

L462: No. The statement is conditioned on precipitation being formed the same way in both simulations, given the different surface fields as input. However, the point of the study is that one simulation has the convection scheme active, while the other one does not.

Please find below an alternative (often discussed) hypothesis that would yield similar differences as those presented:

The soil in simulations typically dries out during the summer months. It is very probable that in autumn soil-water content in the RCM and CPM slightly differ, since soil conductivity, soil diffusivity, evaporation and infiltration rate were not calibrated to yield the exact same soil state. The differences in soil-water content will yield differences in the partitioning of the surface latent and sensible heatfluxes (as observed), and ultimately slight differences in surface specific humidity. The signal appears in the composites of all EOFs because it is already there in the mean.

L465: Where does conclusion (c) come from? Maybe a paragraph went missing?

L479: No. (i) scale-dependency is something different. (ii) The considered indices and case studies are not "extreme". (iii) The manuscript is not assessing how "thermodynamical processes" influence precip. You are looking at composites. (iv) Also consider that "thermodynamical processes" is a rather uncommon term.

L481: I disagree. The analysis also allows other conclusions than portrayed (see above).

L487: It might be worthwhile to note that only daily statistics are considered. In the European summer, a large share of heavy precipitation relates to diurnal convection (see Ban, Kendon). These events only show up when considering hourly precipitation statistics.

L495/L498: I do not agree with the term "explained by" (see above).

L511: Again, there are many other possibilities, e.g., the infamous warm bias related to the reduced turbulent length scale in the CPM simualtions (Baldauf et al., 2011).

**Minor Issues**:

Introduction: The introduction discusses heavy precipitation and the CPM approach. However, context on the employed approach is not provided. E.g., why are "weather types and PCA" insightful and useful tools for the proposed questions?

L80ff: Indeed there is some connection between moisture, moisture flux, instability and precipitation. After all, these aspects have been under investigation for many decades. I think the authors need to be more specific when making their argument. Slso, what "moisture excess" do you refer to? The term suddenly appears out of thin air (maybe a part of a sentence got lost while editing?).

L100ff: Discussion of precipitation under-catch and spatial representativness, in particular for mountainous regions might be worthwhile.

L188: I am skeptic that this dataset qualifies as an "independent" source for validation. I didn't know it before, but the description reads like it is mostly based on other models.

L136: "nominal resolution" Do you mean grid spacing?

L143: It might be worthwhile to note that KLIWA-2.8 is embedded into three nests with 0.44 °, 0.0625 °, and 0.025° grid spacing respectivley, as outlined in Hackenbruch et al., (2016).

L146: Could you confirm that ALP-3 was directly nested into ERA-Interim? I am skeptic because (i) the use of intermediate nests wasnt mentioned for the KLIWA-2.8 simulation either, and (ii) the CCLM-5-0-9-KIT contribution outlined in Coppola et al. (2018) specifically mentions an intermediate nest.

L155: Don't you disregard the lateral relaxation zone + additional margin for spinup. Btw: What upper boundary condition is used?

L173: For d=2, PSI considers the sum of 3 days, right?

L300ff: I am not familiar with the analysis presented in Figure 6. I guess its main purpose is a visualization of the spearman rank correlation, and the mentioned "clustering". Could you elaborate on the argument made using that figure? For now, the text mainly refers to the correlation numbers (0.94, 0.41, 0.48).

L304: Please define "hit rate".

L321: Define "percentage of affected area"

L331: Define "ECOs"

L348: Why not simply write specific humidity and temperature in the figure labels? Same for the remaining figures.

L351: How is "spread" calculated?

Figure 5: "The kernel density at each precipitation intensity is shown by the shaded areas." This shaded area is rather confusing, since the corresponding scale is missing. Also, shouldn't the area be the same size across all presented datasets (i.e., sum up to 1), or is it normalized to one dataset?

What does "maximum grid point precipitation" describe? Is that the highest value ever encountered a any grid point in the analysis domain? If yes, does this man that the data is pooled before computing the boxplot?

Table 3: Define "coverage".

Fig. 10: Why does the term "heavy precipitation" suddenly appear again? How is it related to EOF-1? Also, at what time of the day is CAPE computed?

L50: I do not understand what an "energetic low-level" should be.

L54: Why is Khodayar et al., (2021) cited here? I am not convinced it fits the context very well.

**References:**

Baldauf, M., Seifert, A., Förstner, J., Majewski, D., Raschendorfer, M., & Reinhardt, T. (2011). Operational Convective-Scale Numerical Weather Prediction with the COSMO Model: Description and Sensitivities, Monthly Weather Review, 139(12), 3887-3905.

Schär, C., Ban, N., Fischer, E.M. *et al.* Percentile indices for assessing changes in heavy precipitation events. *Climatic Change* **137,** 201–216 (2016). https://doi.org/10.1007/s10584-016-1669-2

---

## Author Comment (AC1)

**Author reply to RC2 (wcd-2022-11)**

**Major Comment**

*The main aim of this work is the evaluation of COSMO-CLM simulations at different resolutions, to assess the scale-dependency of thermodynamical processes influencing extreme precipitation. This topic is very interesting, since the assessment of very high-resolution climate simulation is a challenging area in the climate community. However, before I can recommend publication, there are some issues that must be addressed, mainly related to formal aspects and not to the scientific content, which is relevant.*

*As a general comment, in some points the English style is poor and must be improved, especially from Section 4 onward. I suggest a general review by a qualified in English support officer.*

We thank the reviewer for the valuable comments and suggestions to improve the manuscript. We acknowledge that the English language can be improved we will revise the language carefully.

*From the title, the reader would expect that general conclusions about regional modelling have been drawn but then, moving throughout the text, he realizes that only one regional model has been considered. As the authors properly say (line 525), the present conclusions cannot be generalized to other regional models. So, I suggest to add "COSMO-CLM" in the title. Moreover, the RCM used (i.e. COSMO) is neither specified in the introduction. Please add a few descriptive lines about this model in the Introduction.*

We will include COSMO CLM in the title. In addition, we will compare our findings with other results from the FPS Convection to draw more generalised conclusions for some aspects. Also the COSMO-CLM model will be presented in the description.

*The captions of many figures are too long and descriptive. I suggest to shorten them and to explain in the caption only what is really shown in the figures, moving the other considerations to the text. For example, in the caption of Figure 10, the sentence "The precipitation days are selected as those over one standard deviation of the Principal Components for EOF-1 shown in Fig. 9" can be removed and included in the main text.*

We agree with the reviewer and follow the suggestion.

*Regarding the conclusions, I think that bullets a, b and c are not so relevant and can be merged into a single bullet.*

Following remarks from other reviewers, the conclusions will be revised. We will take this remark into consideration. Bullets a, b and c will be merged into one.

*I would remove Table 1 and 2, since all the information contained are already provided in the text.*

We thank the reviewer for this suggestion and agree that the observational datasets and our simulations are described in the text. However, we believe Tables 1 and 2 are a good summary for those readers that want to capture information at first glance and a good visual summary of the description in the text.

Moreover, some reviewers focused on this table for relevant information and we would like to keep them as we believe they can be useful.

**Specific comments**

*Line 57: Probably "of" is missing between "development" and "Convection"*

Corrected.

*Lines 90-93: I would avoid using direct questions in an Introduction*

We agree. They will be rephrased as aims of the text.

*Line 94: I suggest to add: "This paper is organized as follows:", before of "In Sect. 2".*

We change it accordingly.

*Line 115: I do not understand the need of comparing HYRAS with ERA5 here, since ERA5 is mentioned here for the first time. If you want to keep this sentence, please add more details about ERA5 and a proper reference.*

The reviewer is correct and we follow the suggestions.

*Line 120: Probably "was" is missing between "MSWEP" and "obtained"*

Corrected.

*Line 125: change "high" with "higher"*

Corrected.

*Line 128: "distance of 250 km". Distance from what?*

Distance between radiosonde launching stations. It has been corrected.

*Line 135 and 283: "recent". Since the period starts in 1961, it is not so recent. Please use another adjective.*

We agree. We now explicitly mention the considered period.

*Line 154: I do not think that you can talk of "small inconsistencies" between the simulations, as the domains are different, the forcing are different, and the resolutions are different.*

We re-write this part and better explain how the high-resolution simulations are used in this specific context. Reviewer 1 also pointed out that these cannot be called "small inconsistencies.

*Line 282-283: It is useless to write the whole name of the model again here. Use only COSMO-CLM.*

The reviewer is correct. We change it.

*Line 293-295: "This not imply…compared to RCM". This sentence is not clear, probably some commas are missing. If I well understand, you claim that CPM performs better than RCM according with literature study. Anyway, your conclusions must be based on the present results and not on literature ones.*

We agree with the reviewer. We will only describe our own results here. Our claim that CPM performs better than RCM is further supported in Sect. 5.1 where we validate specific events using two metrics.

*Line 305-306: "The improvement…in a climatology". There is something wrong in the English.*

We will rephrase this sentence. Now it reads:

*"The improvement shown by CPM with respect to RCM shows the added value of high-resolution in detecting extreme precipitation events in a climatology".*

*Line 307: change "show" with "shows".*

Corrected.

*Line 309: "shows"... "showing"…. Avoid using the verb "show" too many times*

We follow this suggestion.

*Line 316: "al" ?*

Corrected.

*Line 334-335: what do you mean "for its part" ?*

Corrected.

*Line 339: remove comma after 2007.*

Corrected.

*Line 341: change "larg" with "large"*

Corrected.

*Line 363: Avoid using the word "bias" twice*

We follow this suggestion.

*Line 363: The bias is 0.5° also below 925 hPa. Please rephrase.*

We have rephrased this sentence.

*Line 364: "close to 2°". The bias is larger than 2° below 700 hPa.*

We will change the wording and shorten the caption.

*Lines 367-371: This paragraph (and Figure 8) is quite confusing and must be rephrased. It is not clear if you are talking of spatially averaged or temporal averaged bias. Moreover, in the caption of Figure 8, explain better what is shown.*

We will follow the suggestion and revise the pararagraph and the caption of Fig. 8. In any case we are referring to temporally averaged bias. It is the mean bias of all events being validated.

*Line 398-399: probably the verb "explaining" is not correct. Moreover, there is a closed bracket after CPM that was never open.*

We will remove the bracket and rephrase the sentence.

*Figure 9. The caption is not clear. Explain clearly what is shown in each panel (a, b… etc).*

We will revise the caption. Moreover, following reviewer 1 comments' we will revise this section as we will try to fine-tune the method of EOFs and principal components.

*Lines 403-406. It is not clear why red color is referred to CPM and blue to RCM. Probably these maps show the difference between CPM and RCM, but please explain better what is shown and what are you describing.*

The reviewer is correct that the difference is shown (as mentioned in line 403). We will revise the text to clarify it better.

*Line 411-412: "low terrain"? do you mean "low heights"?*

Correct. We will change it, accordingly. We refer to low altitude terrain.

*Line 418: "to be the main precursors of the differences". This sentence is stand-alone and I do not understand what you mean.*

We agree that the wording of the sentence is not clear. Also, by suggestion of Reviewer 1 we are reviewing this aspect for our conclusions. The message of this paragraph is that we believe that the precipitation differences come mainly from the different representation of the dynamic convective processes, e.g., intensification of vertical updraughts and larger triggering of convective cells.

We will rephrase the paragraph to better clarify the findings.

*Line 423: The concept of "preconditioning" is used in numerical analysis. I understand what you mean in this context, but I recommend to use and alternative term here (e.g. pre-existing).*

We will rephrase the sentence. We meant that the differences of the environmental conditions prior to the event. We change it to pre-existing conditions.

*Line 428: "in compared"?  Probably "in" must be removed.*

Corrected.

*Line 431: Explain also here what "theta" represents.*

The parameter "Equivalent Potential Temperature at 850 hPa" was introduced in line 429, but not connected to the symbol. We change that.

*Lines 459-460: "The analysis… processes". The English style is poor.*

We will rephrase the sentence.

*Line 467-468: Again, I believe that "preconditioned" is not the appropriate verb.*

We will revise the sentence to clarify it. We change it for pre-existing conditions.

*Line 471: change "differences" with "different".*

Corrected.

*Line 471: "RCM evaporates more moisture"? RCM is a numerical model and does not evaporate anything.*

The reviewer is of course correct and we will change this wording.

*Line 511: In a similar manner, RCM does not emit sensible heat flux.*

We will change it here, as well.

---

## Author Comment (AC2)

**Author reply to RC3 (wcd-2022-11)**

*The manuscript investigates extreme precipitation events in two sets of regional climate simulations, as well as observational datasets. The focus of the study is on extreme precipitation that was continuously larger than the 80th percentile for at least two days for a given grid point.*

*The RCM simulation was executed at 25 km grid spacing, while the CPM set of simulations was performed at 3 km grid spacing, without parametrization for deep convection.*

*First, the synoptic weather type leading to the extreme precipitation events is determined separately for summer and for winter, then the simulated extreme precipitation is evaluated, followed by an investigation of events, and an investigation of the thermodynamic processes leading to extreme precipitation generation in the two different model configurations.*

*The subject of the manuscript is interesting, the study is performed with adequate techniques, and the presentation and language are of high quality. I thus recommend the publication of the manuscript after minor corrections.*

We thank the reviewer for the valuable comments and suggestions to improve the manuscript.

**General remarks**
*Title: the word "scale-dependency" suggest that the processes are investigated over a continuous range of resolutions, in search for discontinuities. Yet, only two different set ups are presented. Thus, I suggest a renaming of the title to e.g. something like "extreme precipitation processes in regional climate simulations of the greater Alpine Region in convection-permitting and convection-parametrizing simulations".*

We agree with the reviewer. We will change the title considering these comments, also including suggestions from the other reviewers.

*Section 6: scale dependency of thermodynamic processes: a regional weather/climate model forced by boundary data is quite constrained in its way to react, as much of the forcing is provided by the boundary data (as the authors also mention). Thus, part of the analysis in section 6 reveals different strategies of the model configurations to deal with this forcing containing different compensating errors. One forcing mechanism that is not mentioned but that can be of importance, at least for summertime precipitation, is radiative cooling. The radiative cooling leads to a destabilization of the atmosphere, that will enhance convective activity. I suggest to also check the outgoing longwave radiation in the two different sets of simulations for its significance in the extreme precipitation cases.*

We will follow the suggestion and check the long wave radiation. We agree that understanding the differences in longwave radiation between RCM and CPM can complement the finding of the other model variables, especially the surface heat fluxes.

**Specific comments**

*Line 65: numerics and physics-dynamics coupling should also be mentioned.*

We have included this remark.

*Line 293: I disagree with the statement "this does not imply a worse performance by CPM …". The overestimation of grid point extreme precipitation is one of the well-known deficits of convection-permitting models, as you state, despite many advantages. Please reformulate, admitting the issue.*

We have revised the statement and mention this known issue. Now it reads:

*"The comparison against HYRAS-5km (black), shows a good agreement by RCM and CPM for values between 1 mm d$^{-1}$ and 10 mm d$^{-1}$. However, CPM (red) overestimates extreme precipitation for grid point maxima. This is a well-known deficit of CPM (Kendon et al., 2012) in spite of its many advantages e.g., improvements in the representation of the diurnal cycle (Kendon et al., 2012; Lin et al., 2018), or better event-scale representation (Chan et al., 2012; Ban et al., 2018)."*

*Section 5.2: the difference in temperature lapse-rate should be discussed in more detail. The lapse-rate will be the driver for further convective activity. Or formulated differently, the interior of the model domain may take on a different lapse rate in CPM vs CRM to cope with the different representation of convection (compensation model errors again).*

We will consider your suggestion and investigate the lapse-rate differences in more detail.

*Figure 10: some of the effects illustrated are very closely linked together, e.g. the effects seen in near-surface specific humidity and surface latent heat flux.*

We agree that the connection between variables that show relevant resolution effects must be better explained. Some connections are mentioned between lines 459 and 476 but we will extend this information as the reviewer suggests.

For instance, for the relationship between surface specific humidity and latent heat flux emission it is pointed out that:

*"e) the surface specific humidity differences can be 470 explained through differences latent heat fluxes between RCM and CPM, where RCM evaporates more moisture over the Sea and CPM over land."*

**Technical comments**

*Line 59: see also Vergara-Temprado et al., 2020*

We will include a reference to the paper

*Line 140/141: include "of" before "these data sets".*

Corrected.

*Line 156: replace "it" by "they"*

In this case we are referring to just one area, the SGer area.

*Line 221: include "of" between "range" and "values"*

Corrected.

*Line 241: should "flowing" be "following" ?*

Corrected.

*Line 282: OSMO → COSMO*

Corrected.

*Line 305: remove "is"*

Corrected.

*Line 316. al → all*

Corrected.

*Legend Figure 6: insert space before "HYRAS"*

Corrected.

*Line 305: remove "is"*

Corrected.

*Line 316. al → all*

---

## Author Comment (AC3)

**Author reply to RC1 (wcd-2022-11)**

**General Comments**

*The presented manuscript assesses differences in the statistics of daily ("heavy") precipitation between a 25km RCM simulation and two kilometer-resolution CPM simulations. This analysis has been done before, the results are quite robust across model codes and well described in the literature. The presented analysis is novel nevertheless, but resorts to rather sophisticated methods (PCA, composites) and indices (PSI, FSS, wet-day percentiles), making them rather complex and difficult to interpret. Unfortunately, some of their implementation details are not fully fit for purpose (see major issues below).*

We thank the reviewer for the very enlightening and thorough comments of her/his report. In the following we provide our answers to the general major and minor comments as well as implemented changes in the manuscript. We hope that these changes and our answers, improve our paper to better meet the journal standards.

Regarding the reviewer's comments, we acknowledge that the combination of different statistical and computational techniques (PCA, composites, PSI, FSS, wet-day percentiles), have made our manuscript complex and less readable. We will revise our manuscript to improve the readability and to better describe these techniques, their purposes, and results. In particular, we considered the PCA and composites analyses suitable because they are a good means to extract relevant modes of precipitation events. The advantage with respect to e.g., individual case assessment is that they allow to combine several "similar" precipitation situations into one mode. Regarding the PSI and wet-day percentiles, we were interested in studying the capabilities of the PSI, that has the advantage of allowing a flexible definition of precipitation threshold and persistence. We thought, the PSI could bring added value in comparison to other simpler indices. Finally, we chose the FSS as a score allowing for validation of precipitation structures in modelled simulations, where we have had good results in previous studies, e.g., (Caldas-Alvarez et al., 2021).

*Also, I am not convinced that the presented results allow supporting the interpretations and conclusions made at the end of the manuscript. In particular, I criticize some of the inferences made from the detected differences (see major issues below), albeit alternative hypothesis are discussed in the literature. Finally, the detected differences are rather small, while the necessary statistical quantification is not provided (E.g., L490 and L303). In my role as a reviewer, I usually abstain from requiring stat. tests, but when an established and robust hypothesis (no differences in daily statistics) is refuted in a manuscript, requiring a robustness assessment seems warranted.*

We admit that some of the conclusions were either not well formulated or not optimally underpinned. We will reformulate these statements and improve the reasoning, including alternative hypothesis. See specific answers below.

*I have the impression that the manuscript title does not reflect the content of the manuscript precisely enough, as neither "scale-dependency" nor "extreme precipitation processes" are actually assessed in the study.*

Regarding the title, we will modify it, following the reviewer's advice to better address the content and results of our work. Also other reviewers commented on the title so we will correct it to meet the reviewers' suggestions.

*In particular, I don't think that the chosen indices and events qualify as "extreme precipitation", and "extreme precipitation processes" are not considered at all. Meanwhile, the term "scale-dependency" is usually used differently from the presented use case.*

We agree that some of the analysed events cannot be considered extreme precipitation events. We have adapted the terminology to "heavy precipitation" which we believe is more suitable. Other papers in the field have denominated heavy precipitation events those above 100 mm/d as is the case for those analysed in the HyMeX consortium (Khodayar et al., 2021). This precipitation amount fits better the range of intensities covered in our paper. Furthermore, the index used for event identification (PSI), relies on spatial percentiles implying that locations over dry areas need lower precipitation totals to be highlighted as "rare" or extreme. Hence, some events could be detected with precipitation below 100 mm/d if they happened over typically dry areas (more on this in the specific answers below). To summarize, we agree that the precipitation ranges considered here are not extreme. We believe a better term for those cases is "heavy precipitation". If events with totals below 100 mm/d are detected and mentioned we will specifically refer to them as moderate precipitation events.

Although this will be developed further later in this document, we agree that there is a problem with our approach to "extreme precipitation processes". Model variables other than precipitation, that have an influence on the simulation of the analysed events are studied in our manuscript using composites. Our aim is to highlight the differences in the model variables affecting precipitation between the two configurations RCM (25 km) vs. CPM (3 km). We agree that this is not the same as studying "heavy precipitation processes". Hence, we will refer to differences in model variables between our RCM and CPM simulations during the heavy precipitation days. Furthermore, we will reformulate our PCA and composite analyses to work only with heavy precipitation days, as opposite to the previous version where "all days" were considered. Instead of using "all days" we will derive our PCA and composites from precipitation days in the upper percentiles of the distribution.

Also, regarding terminology, we adopted the term "scale-dependency" to discuss model differences due to resolution (and their corresponding configurations for our climate model). This term has been used in this same sense in other papers of the field (e.g., Helsen et al., 2020; Glotfelti et al., 2020; Tölle et al., 2020).

*Overall, the simulation configurations are not described in sufficient detail and some important aspects are missing. E.g., for the ALPS-3 simulation, the authors mostly refer to Coppola et al.,(2018), which is a MIP overview paper, and thus does provide the necessary detail to ensure scientific reproducibility. For instance, the info that KLIWA-2.8 and ALPS-3 were conducted using two different major releases of the code was not highlighted. I consider this detail is relevant context when suggesting that ALPS-3 is merely a continuation of KLIWA-2.8.*

The reviewer is of course correct. The setup for both simulations differs in many details, although there are as well similarities like resolution, overlapping domain and the basic characteristics of CPM simulations with CCLM. Therefore, we applied it for the specific purpose of identifying heavy precipitation events over an extended period (Fig. 6). The reviewer mentions, the KLIWA ensemble is just used in chapter 4. We will clarify the approach in several ways (see also specific comments): 1) Improved description of the simulations, 2.) Better clarify the specific purpose (see answer to major comment 2). 3. Change figure 5 and 4.) indicate the temporal separation between the "KLIWA period" and the "ALP-3 period"

**Major Issues**

*L153: "In spite of these small inconsistencies, we combine both CPM simulations to attain a sufficiently large investigation period for comparison with the RCM simulation and observational data sets" I don't think that the term "small inconsistencies" is justified here. The KLIWA-2.8 and ALP-3 configurations differ in virtually every aspect, apart from their overlapping computational domains and their use of the COSMO code (in two different major releases)*

We agree that the formulation is not adequate and will correct that. We will rewrite the model description section and be more specific, as was requested by the reviewer. It was not meant in the sense, that they can be used as one homogeneous data set. In fact, the paper heavily relies on the ALP-3 data and KLIWA2.8 is only used for a specific purpose in section 4 (see reply to the next question).

*Also, what defines " sufficiently large" KLIWA-2.8 is 29 years long. I'd consider that sufficient for all of the presented analysis and the qualitative conclusions of the study. Note that KLIWA2.8 is actually only used in Section 4*

We agree that the wording of this sentence is incorrect. We correct it in the newer version of the manuscript.

As the reviewer mentions, KLIWA is only used in section 4 and Figs 5 and 6. We decided to include it to have a longer period for the event detection. We believe this is useful because if we only focus on the e.g., top 500 events, in a shorter period this would correspond to one event per week. On the other hand, combining KLIWA and ALP-3 allows us to cover a 45-year period and the TOP 500 events would correspond to one event per month.

On another note, both simulations are at least comparable in the grid resolution and provide an overlapping area, which seems to use sufficient for the purpose. We will clarify this in the text and indicate the separation between to two datasets in Figure 6. We will also replace Figure 5 and include information separately for ALP-3 and KLIWA2.8, which will enable a good comparison of the two datasets

*L170 Does the 80th all-day-percentile really characterize "high grid-point intensity"? I think the 80th percentile represents a few mm/days, which is rather typical for a rainy day in Germany*

We agree that the 80th all-day percentile corresponds to a few mm/days. We will correct this in the manuscript. The selection of heavy precipitation events comes in the second step of the process where, after having calculated a PSI value for each day, we retain only events in the upper 90th, 95th percentile of the PSI distribution.

*L193: I don't understand in Fig 2 the 11th June event is below the 90th percentile of both respective indices, no? This is exactly opposite to the argument made in the text.*

Thank you for pointing the mistake out, we correct this part of the analysis (see next question).

*L198: I don't think that the analysis supports this claim, rather to opposite. As indicated in Fig.2 the Spearman's rank correlation between PSI and fldsum is 0.98/0.96. Also it is evident that the solid line mostly tracks the dotted line. That is, at lower amplitude, which does not matter for a ranked index. This means that after applying the second threshold (i.e. L236, L301) almost the exact same events are chosen as would have been when using the fldsum. In other words, Fig.2 actually demonstrates that PSI is an unnecessarily complex choice for the presented use-case.*

Figure 2 is a bad choice to show the added value of the PSI. We modify this analysis and graphs in the manuscript. Instead, we focus on the rank correlations between *fldsum* and PSI and include a new discussion of the PSI capabilities.

The main points of our new analysis are summarized in the following:

- PSI performs similarly to *fldsum* when we choose a low percentile threshold and zero days for the $RR_{perc_{ij}}$ and $d$ parameters, respectively. As the reviewer points out "the solid line mostly tracks the dotted line" in such a configuration (for instance with $RR_{perc_{ij}} = RR_{80_{ij}}$ and $d = 0$).
- The PSI performs differently to *fldsum* if $RR_{perc_{ij}}$ and $d$ are set to a higher threshold and $d \neq 0$. In fact different events are detected and the rank correlation with *fldsum* is somewhat lower (0.86) with $RR_{perc_{ij}} = RR_{95_{ij}}$ and $d = 2$.
- Different events are detected, because: a) persistence plays a role (an HPE lasting for 2 days will be preferred to a 1-day HPE of the same intensity); b) "rarer" events are preferred (events occurring over a dry areas or a very heavy events, to surpass the 80[th], 95[th] percentile of precipitation over a grid point).
- A different application of the PSI is suggested where instead of using a percentile to define the threshold ($RR_{perc_{ij}}$), an absolute value could be chosen, e.g., 100 mm/d. In this case only grid points with precipitation larger than this value are included in the calculation.
- PSI has larger flexibility for extremes detection than fldsum due to the fine-tuning of the aforementioned parameters.

To show these points we are preparing a new table of ranked events detected with fldsum and different combination of PSI settings. We will include spatial distributions to illustrate how the PSI detects different events than the fldsum which could be more useful to the needs of a potential user of the PSI.

*L240: Why are the EOFs computed using the RCM and not ERA-Interim directly? My point is that the first few EOFs of the 500 hPa geopotential need to be almost exactly the same in RCM and driving data. After all, if those EOFs of the RCM simulation would become systematically different than those in the driving data, the RCM approach would become somewhat questionable.*

We agree with the reviewer. This was exactly our assumption to derive EOFs of geopotential height at 500 hPa from the RCM simulation. As the reviewer says, the first EOFs will not be substantially different between RCM and the driving model as no relevant differences are expected between these two simulations for geopotential height at 500 hPa. Hence, we decided to proceed with RCM.

*L285: The analysis presented in Figure 5 is rather difficult to interpret, since the boxplot parameters (median, quartiles, ...) depict percentiles of a conditional index (i.e., the wet-day precip., P > 1 mm/day). Please consult the following study for a thorough discussion of the problems related to deriving percentiles of conditional indices: Schär et al., (2016)*

Following the indications of the reviewer and the conclusions of Schär et al., (2016) we will modify the study of percentile differences between RCM and CPM to consider an all-day index. We agree that using a wet-day index, to obtain conclusions for percentile variations (between RCM and CPM or observations) can be misleading. Instead an all-day index guarantees that the same absolute probabilities are used for comparison.

*L327: Why are events chosen that entail bad quality/non-existent observations (since over ocean)? To verify a model, I would intuitively choose case studies that have abundant high-quality observations available. Also, I am not fully convinced about the usefulness of the MSWEP-11km product.*

We decided to validate our simulations against MSWEP-11km to also profit from their coverage over the Mediterranean. We did not select events that specifically affect the Mediterranean Sea, but events which were heavy and were representative of the area analysed (greater Alpine area) and the two seasons (summer and winter). We considered the coverage of water surfaces an advantage of MSWEP-11km with respect to HYRAS and EOBS.

We also opted for MSWEP-11km because they contain station data from the Climate Prediction Center (CPC) and the Global Precipitation Climatology Centre (GPCC), in addition to remote sensing, e.g., PERSIANN, TRMM. Furthermore, we chose MSWEP-11km because of its good performance in previous evaluations against station data, globally (Beck et al., 2017, 2019; Xiang et al., 2021) and over specific geographies (Du et al., 2022; Peña-Guerrero et al., 2022). In this sense the station data included in MSWEP from the CPC and the GPCC, have a good coverage over Europe, hence we believe this product to be a good choice for our event validation.

These reasons are emphasized in the new version of the manuscript.

*"We use the MSWEP product to profit from its high accuracy, shown in previous studies, globally (Beck et al., 2017, 2019; Xiang et al., 2021) as well as in specific geographies (Du et al., 2022; Peña-Guerrero et al., 2022). MSWEP has the advantage of covering sea surfaces and is adequate for precipitation event evaluation because it includes station data from CPC and GPCC."*

*L361: The statement needs to be qualified, also w.r.t. internal variability and accuracy of specific humidity obs with radiosondes. In fact, I was surprised to see only a difference of 0.1 — 0.2 g/kg when comparing profiles of a limited-area climate simulation to more-or-less "instantaneous" soundings. I think, my view is corroborated well when considering Fig. S2, instead of the differences. In contrast to the authors, I think these results are actually rather promising.*

We will revise our conclusions from this analysis to better assess the performance of the RCM and CPM simulations.

We acknowledge that we overstated the incapability of COSMO-CLM to represent the temperature and humidity profiles and agree with the reviewer that a deviation of 0.1 to 0.2 g/kg is actually a promising result. We will rephrase our conclusions in this paragraph to better provide this information.

Our point is that humidity deviations of less than 1 g/kg in the lower troposphere can have relevant implications for precipitation representation in our RCM and CPM set-ups. This was actually shown in previous publications in our working group (Caldas-Alvarez and Khodayar, 2020; Caldas-Alvarez et al., 2021) for two HPEs and in other sensitivity studies (Honda and Kawano, 2015 and Lee et al., 2018).

Here, we cannot assert whether the observed differences in the humidity profiles between RCM and CPM have an influence on precipitation differences as only 8 cases are considered, and this model sensitivity is not isolated. However, we include this analysis to provide the reader with useful information about the magnitude of the model biases with respect to observational soundings. We also think this comparison against observations is useful since later in the paper humidity and temperature differences between RCM and CPM will be discussed.

To conclude, we will revise this analysis to provide a fairer description of the models' performance and to provide an estimation of the magnitude of the model biases with respect to observations.

*L384ff: I am not completely sure if the chosen procedure is appropriate, but honestly, I do not fully understand why it has been chosen in the first place. First of all, why would EOF1 be associated with "heavy precipitation"? I thought that EOF1 portrays the mode with the largest variance, right? That is (by definition) rather unlikely a percentile at the tail of the precip. Distribution, no? Second, EOFs seems rather complicated provided that the results (Fig. 9) exert a very similar pattern as much simpler indices, e.g., the standard deviation (cdo timstd).*

We totally agree with the reviewer. We are correcting this in the revised version of the manuscript. The error is in one of our departing assumptions. In the first version of the manuscript we used "PCA as described in Sect. 2.3.2 with daily precipitation in the period 2000-2015 from RCM and CPM." As the reviewer points out, the leading mode of precipitation is the most common pattern, not heavy precipitation.

To avoid this issue, we correct our approach to obtain the PCs from heavy precipitation days exclusively. We filter, by means of the PSI, days with heavy precipitation in the period (above the upper 90th percentile) and obtain the PCs are obtained from that sample. This is on-going work that will be presented in the new version of the manuscript.

*L391: Why for the day prior? I thought the authors suggest the detected heavy precipitation events to be primarily associated with synoptic systems.*

We are interested, foremost, in differences in the ground and local conditions (soil temperature and fluxes, cape, surface winds) between RCM (25 km) and CPM (3 km) that could explain the final differences in precipitation amount (in the composites). Selecting the day prior, we try to observe those differences when precipitation has not yet started, or when precipitation is not yet heavy. This is especially relevant for moisture transport that builds-up on the hours prior to precipitation initiation. Our goal is to find out whether changes due to model resolution exist in the pre-conditioning of heavy precipitation are related to precipitation differences.

As the reviewer points out differences in these variables will be most impacting for cases with a weak synoptic forcing. This is more typical of summer events as in winter events.

We have tested both approaches and could not find substantial differences, hence we would continue showing composites of the day prior to profit from the advantage of understanding model differences during the moisture build up.

*L398/399: If the first three leading EOFs only explain 39% of the variance. Is that analysis really an appropriate tool? Maybe I do not understand precisely what the authors are trying to achieve.*

We expect to explain a larger amount of precipitation variance with the new implementation of the method, i.e., extracting the EOFs from heavy precipitation days exclusively (see previous questions).

We believe the EOFs approach is suitable for our purposes since it helps us encompassing several heavy precipitation days into the single EOFs. By doing so, we can study model differences on a lower number of precipitation spatial patterns.

*L415ff: How come? Why not a parameter choice, or a time-step sensitivity in the cloud microphysics parameterization, or the nesting strategy ...*

We agree that our presenting of conclusions at this part has been too categorical. We cannot, and should not, conclude that precipitation differences between RCM and CPM being are due to

differences in vertical wind speeds and more triggering of convective cells only. The dominating factors for the differences are difficult to disentangle provided the large number of differences in the modelling configurations, e.g., parameter choice, schemes, etc. However, we believe the differences in vertical wind speeds are important to explain these differences, based on previous publications in the field (Langhans et al., 2012; Barthlott and Hoose, 2015)

We have adapted the wording in the same paragraph to include this remark.

*"The findings based on the main modes of precipitation variance, for which EOF-1 is shown as an example, can be summarized as follows: (a) CPM displays larger precipitation than RCM over the mountains for all assessed EOFs and winter and summer seasons. Resolution differences in dynamic processes, e.g., increased vertical wind speeds, larger triggering of convective cells (Langhans et al., 2012; Barthlott and Hoose, 2015) could play an important role in these differences, invigorating the precipitating systems for this mode especially since EOF-1 has a marked orographic pattern."*

*L462: No. The statement is conditioned on precipitation being formed the same way in both simulations, given the different surface fields as input. However, the point of the study is that one simulation has the convection scheme active, while the other one does not.*

We agree that our conclusions were too categorical (see previous question). While resolution differences in the analysed model variables exist on the day prior to precipitation and are non-negligible, (e.g., surface specific humidity 9shows differences up to 1.2 mm) we agree that we cannot conclude they are responsible for changes in precipitation. As the reviewer points out, several other changes in the model configurations of RCM and CPM could be responsible for the precipitation differences.

We will then focus in assessing the differences in the model variables without implying causality for precipitation differences. We will likewise provide plausible explanations for the model differences without overstating our findings.

We believe nevertheless that it is relevant for the climate modelling community and potential readers of our paper knowing about how the RCM and CPM show differences in our particular simulations with CCLM.

*Please find below an alternative (often discussed) hypothesis that would yield similar differences as those presented:*

*The soil in simulations typically dries out during the summer months. It is very probable that in autumn soil-water content in the RCM and CPM slightly differ, since soil conductivity, soil diffusivity, evaporation and infiltration rate were not calibrated to yield the exact same soil state. The differences in soil-water content will yield differences in the partitioning of the surface latent and sensible heat fluxes (as observed), and ultimately slight differences in surface specific humidity. The signal appears in the composites of all EOFs because it is already there in the mean.*

We agree that it was not precisely enough formulated and will provide a more detailed discussion, including the effects of the small-scale heterogeneities and the differences between parametrized and not parametrized precipitation. See also answers to the point above. We will consider this explanation in our new version of the manuscript to explain soil moisture and surface heat fluxes differences.

*L465: Where does conclusion (c) come from? Maybe a paragraph went missing?*

Conclusion c) comes from the study of the remainder EOFs. These are provided in the supplementary material due to the extension of the paper.

However, since the EOFs will be processed and obtain again, using only heavy precipitation days, this paragraph will be analysed again in depth.

*L479: No. (i) scale-dependency is something different. (ii) The considered indices and case studies are not "extreme". (iii) The manuscript is not assessing how "thermodynamical processes" influence precip. You are looking at composites. (iv) Also consider that "thermodynamical processes" is a rather uncommon term.*

We will work on the wording of this paragraph to convey the concerns pointed out here. Regarding the term "extreme" precipitation, this will be changed for "heavy" following the new approach for EOF derivation explained before (see previous questions). Likewise, we will not talk about thermodynamic processes as the reviewer suggest, but rather the specific model variables will be mentioned. Finally, scale-dependency refers to modelling studies of simulations with different resolutions. This term has been used in previous literature of the like, e.g., field (e.g., Helsen et al., 2020; Glotfelti et al., 2020; Tölle et al., 2020).

*L481: I disagree. The analysis also allows other conclusions than portrayed (see above).*

We have corrected this in the new version of the manuscript (see above).

*L487: It might be worthwhile to note that only daily statistics are considered. In the European summer, a large share of heavy precipitation relates to diurnal convection (see Ban, Kendon). These events only show up when considering hourly precipitation statistics.*

We agree with the reviewer and specify that daily precipitation statistics are used, to avoid confusion with sub-daily temporal scales. The paragraph now reads:

*"Using daily precipitation data we find that summer events are associated to either frontal convection on the western sector of elongated upper-level troughs and evolved cut-off lows (EOFs 1, 3 and 4), or due to winter-like synoptic patterns of stationary fronts over central Europe or strong zonal flows (EOFs 2 and 5). Five PCs are sufficient to explain the major part of the natural variability of summer cases"*

*L495/L498: I do not agree with the term "explained by" (see above).*

We agree with the reviewer that resolution differences in the dynamic processes describe do not "explain" alone the precipitation differences. As stated, before we will change the text to better convey this information. The changes in the number of convective cells triggered or the vertical wind speeds are two possible precursors that we consider relevant but we agree that we cannot states these "explain" the changes in precipitation over the mountain.

**Minor Issues**

*Introduction: The introduction discusses heavy precipitation and the CPM approach. However, context on the employed approach is not provided. E.g., why are "weather types and PCA" insightful and useful tools for the proposed questions?*

We will include this information in the introduction of the manuscript. We believe our approach using weather types and PCA is useful because it helps us extract the predominating spatial distributions of atmospheric modelling variables, e.g., 500 hPa geopotential and precipitation. By doing so we can comprehend several thousand days of decadal data into a few statistically differentiated groups where we can observe the modelling differences of our two configurations RCM and CPM.

*L80ff: Indeed there is some connection between moisture, moisture flux, instability and precipitation. After all, these aspects have been under investigation for many decades. I think the authors need to be more specific when making their argument. Also, what "moisture excess" do you refer to? The term suddenly appears out of thin air (maybe a part of a sentence got lost while editing?).*

Our argument is that it is worthy investigating what the implications of the moisture wet biases in RCM and CPM are for precipitation representation. Our paper touches, to some extent, upon this topic as surface humidity biases, and moisture flux differences in both configurations (RCM and CPM) are analysed.

"Moisture excess" means moisture wet bias, but we agree that this term can be misleading and we change it for moisture wet bias.

*L100ff: Discussion of precipitation under-catch and spatial representativeness, in particular for mountainous regions might be worthwhile.*

In our manuscript we do not describe the implications for precipitation under-catch so we believe it would be better not to introduce this topic in the manuscript. Regarding spatial representativeness over mountain terrain we will briefly introduce the implications and differences between RCM and CPM.

*L188: I am skeptic that this dataset qualifies as an "independent" source for validation. I didn't know it before, but the description reads like it is mostly based on other models.*

We are not sure, what the reviewer means here. The paragraph described the effect of different parameters in the PSI calculation on the results. The examples are taken from the HYRAS gridded observation and are not meant as a model evaluation.

*L136: "nominal resolution" Do you mean grid spacing?*

Correct. We meant the grid spacing here and will change the phrasing

*L143: It might be worthwhile to note that KLIWA-2.8 is embedded into three nests with 0.44 °,0.0625 °, and 0.025° grid spacing respectively, as outlined in Hackenbruch et al., (2016).*

We will describe the nesting strategy better in the restructured chapter 2.2

*L146: Could you confirm that ALP-3 was directly nested into ERA-Interim? I am skeptic because (i) the use of intermediate nests wasn't mentioned for the KLIWA-2.8 simulation either, and (ii) the CCLM-5-0-9-KIT contribution outlined in Coppola et al. (2018) specifically mentions an intermediate nest.*

The description in Coppola et al. (2018) is correct. The ALP-3 km simulation was nested within the EUR-22 simulation used here. We will clarify that better in Chapter 2.2, which will be rewritten anyway

*L155: Don't you disregard the lateral relaxation zone + additional margin for spinup. Btw: What upper boundary condition is used?*

We excluded the lateral relaxation zones and the additional margin for the spin-up of 23 grid boxes or about 70 km on each boundary for ALP-3 and 11 grid boxes for the EUR-22 domain (~275 km). CCLM uses Raleigh-damping at the upper boundary above 11.000 m which is the default setup for a European domain. We will include the information in Chapter 2.2

*L173: For d=2, PSI considers the sum of 3 days, right?*

Yes.

*L300ff: I am not familiar with the analysis presented in Figure 6. I guess its main purpose is a visualization of the spearman rank correlation, and the mentioned "clustering". Could you elaborate on the argument made using that figure? For now, the text mainly refers to the correlation numbers (0.94, 0.41, 0.48).*

Yes, a dot diagram is used to represent a series of events in a time span. The figure aids visualizing which events are represented in the model or found in the observations.

*L304: Please define "hit rate".*

We include a brief definition.

*L321: Define "percentage of affected area"*

Ok.

*L331: Define "ECOs"*

Is defined in line 260.

*"Elongated Cut-Off (ECO)"*

*L348: Why not simply write specific humidity and temperature in the figure labels? Same for the remaining figures.*

We will incorporate this suggestion in the labels.

*L351: How is "spread" calculated?*

It is the standard deviation of the differences.

*Figure 5: "The kernel density at each precipitation intensity is shown by the shaded areas." This shaded area is rather confusing, since the corresponding scale is missing. Also, shouldn't the area be the same size across all presented datasets (i.e., sum up to 1), or is it normalized to one dataset?*

It is normalized to each data set.

*What does "maximum grid point precipitation" describe? Is that the highest value ever encountered a any grid point in the analysis domain? If yes, does this mean that the data is pooled before computing the boxplot?*

It is the maximum grid point precipitation of the analysis domain, represented by each simulation or observations) for the full period 1971 to 2015.

To derive the box plots, we use all daily grid point precipitation data (observations and simulations).

*Table 3: Define "coverage".*

Coverage is the percentage of affected area above the 80th percentile. We include this information in the text.

*Fig. 10: Why does the term "heavy precipitation" suddenly appear again? How is it related to EOF-1? Also, at what time of the day is CAPE computed?*

We will change the term to heavy precipitation throughout the whole text in the revised version of the manuscript (see above). Moreover, with the new approach for EOF and PC derivation employing

heavy precipitation days, EOF-1 will be the first mode of heavy precipitation variance. CAPE is computed on the day prior to heavy precipitation.

*L50: I do not understand what an "energetic low-level" should be.*

It has been corrected to "warm and moist" low level.

*L54: Why is Khodayar et al., (2021) cited here? I am not convinced it fits the context very well.*

We take this reference out.

**References**

Barthlott, C. and Hoose, C.: Spatial and temporal variability of clouds and precipitation over Germany: multiscale simulations 705 across the "gray zone", Atmospheric Chemistry and Physics, 15, 12 361–12 384, https://doi.org/10.5194/acp-15-12361-2015, 2015.

Beck, H. E., Pan, M., Roy, T., Weedon, G. P., Pappenberger, F., van Dijk, A. I. J. M., Huffman, G. J., Adler, R. F., and Wood, E. F.: Daily evaluation of 26 precipitation datasets using Stage-IV gauge-radar data for the CONUS, Hydrology and Earth System Sciences, 23, 207–224, 2019.

Beck, H. E., van Dijk, A. I. J. M., Levizzani, V., Schellekens, J., Miralles, D. G., Martens, B., and de Roo, A.: MSWEP: 3-hourly 0.25global gridded precipitation (1979–2015) by merging gauge, satellite, and reanalysis data, Hydrology and Earth System Sciences, 21, 589–615, 2017.

Du, Y., Wang, D., Zhu, J., Lin, Z., and Zhong, Y.: Intercomparison of multiple high-resolution precipitation products over China: Climatology and extremes, Atmospheric Research, 278, 106342, 2022.

Glotfelty, T., Alapaty, K., He, J., Hawbecker, P., Song, X., and Zhang, G.: Studying Scale Dependency of Aerosol–Cloud Interactions Using Multiscale Cloud Formulations, Journal of the Atmospheric Sciences, 77, 3847–3868, 2020.

Helsen, S., van Lipzig, N. P. M., Demuzere, M., Broucke, S. V., Caluwaerts, S., Cruz, L. D., Troch, R. D., Hamdi, R., Termonia, P., Schaeybroeck, B. V., and Wouters, H.: Consistent scale-dependency of future increases in hourly extreme precipitation in two convection-permitting climate models, Climate Dynamics, 54, 1267–1280, 2019.

Khodayar, S., Davolio, S., Di Girolamo, P., Lebeaupin Brossier, C., Flaounas, E., Fourrie, N., Lee, K.-O., Ricard, D., Vie, B., Bouttier, F., Caldas-Alvarez, A., and Ducrocq, V.: Overview towards improved understanding of the mechanisms leading to heavy precipitation in the western Mediterranean: lessons learned from HyMeX, Atmos. Chem. Phys., 21, 17051–17078, https://doi.org/10.5194/acp-21-17051-2021, 2021.Langhans, W., Schmidli, J., and Schär, C.: Mesoscale Impacts of Explicit Numerical Diffusion in a Convection-Permitting Model, Monthly Weather Review, 140, 226–244, https://doi.org/10.1175/2011mwr3650.1, 2012.

Peña-Guerrero, M. D., Umirbekov, A., Tarasova, L., and Müller, D.: Comparing the performance of high-resolution global precipitation products across topographic and climatic gradients of Central Asia, International Journal of Climatology, 42, 5554–5569, 2022.

Tölle, M. H., Schefczyk, L., and Gutjahr, O.: Scale dependency of regional climate modeling of current and future climate extremes in Germany, Theoretical and Applied Climatology, 134, 829–848, 2017.

Xiang, Y., Chen, J., Li, L., Peng, T., and Yin, Z.: Evaluation of Eight Global Precipitation Datasets in Hydrological Modeling, Remote Sensing, 13, 2831, 2021.

---

## Referee Report (RR1)

**Anonymous Review of the manuscript "Regional Climate and Convection-Permitting Modelling of heavy precipitation in decadal simulations of the greater Alpine region with COSMO-CLM" by Caldas-Alvarez et al.**

The manuscript has changed substantially since the initial submission, and obviously a lot of work went into it. I appreciate that my extensive comments from the last review round were carefully addressed. I have a few remaining issues that, whilst still important, can be solved rather quickly.

**Major Comments**

I do not fully agree with the statement about the overestimation of precipitation over the Alps (L22/L542). For this argument, the authors primarily refer to studies that apparently support their claims. However, the authors of these studies are more careful about that statement, and typically refer to uncertainties related to the gridding procedure, sampling biases due to the gauges being primarily located in valleys, and the prominent under-catch issue by gauges during HP events. I agree with these studies that, while generally useful, the current datasets/observations are not fit for making the "HP overestimation in CPMs" claim in the Alps.

I am also a bit concerned about the discussion of IWV and its relation to HP (L30 - 35 and L565 – 587). In particular, I did not fully comprehend the presented argument. That is because the authors primarily present results, rather than their idea, and leave most of the interpretation to the reader. It would be helpful if the assessed hypothesis were concisely stated in the introduction, and then assessed in Chapter 8. I currently interpret the results as if the authors alluded to IWV (and remote sensible and latent fluxes over the ocean) being an important driver for differences in HP between RCM and CPM. I don't think that the presented analysis would convincingly outline how such a mechanism would work.

**Minor Comments**

L75ff: The argument presented by Hohenegger et al. (2009) has meanwhile been augmented and better understood. In particular, the sentence starting on L77 is now outdated. More recent hypothesis actually involve a spatial scale dependency ;-)  (e.g., Taylor et al. 2012), that is actually represented in kilometer-resolution climate simulations (e.g., Leutwyler et al. 2021).

L181/L252: In the revised version of the manuscript the 80[th] all-day percentile is used, right? Maybe thus mention "all-day" explicitly in the text?

L303: The text reads as if the authors are talking about the probability of exceedance. Why not show this metric instead (Figure 5)?

Section 6/Figure 9: The authors explain what they did, but I do not fully comprehend what they want to demonstrate. For unfamiliar readers (like me), it might be worthwhile to add a sentence or two explaining the intent at the beginning of the Section (same for L440ff), and a few words at the end summarizing the findings.

L485: I do not understand the role of the green shading. Is there something awkward with "negative" sensible heat fluxes? I think the authors need to explain better what the problem is. Also, I am not familiar with the term "surface directed fluxes" used in the caption of Fig 12, and I do not immediately grasp what it means. Please explain it in the caption.

L497 – L507: I think these paragraphs need to be rewritten. They read like notes rather than actual paragraphs, also I am confused what the underpinning message is.

L572: "overestimates" and "overestimation". I think these words can only be used when comparing against observations. What is wrong with the word "larger"?

L576: "The wind transports the moisture excess in RCM inland." How do you know that? I think such a statement would require a moisture budget, possibly even including a trajectory analysis.

L595: Maybe the original studies merit citation instead? Chubb et al., (2015) provide a nice summary.

**Suggestions**

Title: "Regional Climate and Convection-Permitting Modelling of heavy precipitation ..." Maybe better: Convection-Parameterizing and Convection-Permitting Simulations of Heavy precipitation […]. The term RCM usually refers to the limited-area extent of the computational domain rather than to the parameterization of convection.

L97: Maybe add a sentence outlining how the method will be exploited in the presented study?

L335: Maybe explain what you mean with "event scale"?

L404ff: One might summarize the surface T and QV verification by stating that the event scale verification is consistent with the well-known too dry too hot bias of CPMs. In particular because most of the assessed events are in summer.

L566: I thought you wanted to remove the term "moisture excess" from the manuscript.

**References**

Chubb, T., Manton, M. J., Siems, S. T., Peace, A. D., & Bilish, S. P. (2015). Estimation of Wind-Induced Losses from a Precipitation Gauge Network in the Australian Snowy Mountains, *Journal of Hydrometeorology*, *16*(6), 2619-2638.

Leutwyler, D., Imamovic, A. and Schär, C. (2021). The Continental-Scale Soil Moisture–Precipitation Feedback in Europe with Parameterized and Explicit Convection, *Journal of Climate*, *34*(13), 5303-5320.

Taylor, C. M., R. A. M. de Jeu, F. Guichard, P. P. Harris, and W. A. Dorigo, 2012: Afternoon rain more likely over drier soils. *Nature*, **489**, 423–426.

---

## Author Response (AR2)

**Author reply to the Editor (wcd-2022-11)**

We thank the editor, D. Domeisen, for the useful comments and revision of our manuscript. We are happy to have selected Weather and Climate Dynamics as we feel the subject of our manuscript fits the scope of this journal and its special issue on past and future European atmospheric extreme events under climate change.

In the following, we provide a point-by-point answer to the editor's comments to be added to our answers to the reviewers to complement the revised version of the manuscript and the corresponding track-changes version.

**Editor Comments**

*In the first round of reviews, reviewer three asked about the scale dependency of thermodynamic processes and suggested to look into outgoing longwave radiation. I don't see this comment addressed in section 6 of your revised version, could you please make sure to include it and clarify?*

The results and discussion of our analysis on Outgoing Longwave Radiation (OLWR) is included in Sect. 7.3 (Soil-atmosphere interactions).

We applied the composite analysis method to OLWR as suggested by reviewer 3, and the results are shown in Figs. 12 and 13 and Figs. S14 and S15 of the SM.

Overall, the composite analysis showed that OLWR differences are linked to surface temperature differences between RCM and CPM, irrespective of the analysed precipitation EOF or season. That is why, generally, OLWR is larger in RCM than CPM (up to 9 W m$^{-2}$). However, for summer events over the Po Valley these differences can disappear or even be larger in CPM in agreement with the warmer-drier amplification shown by CPM over this area in our study and Sangelantoni et al., (2020) during heat waves.

This information is conveyed twice in Sect. 7.3 for Winter and Summer events:

*"Finally, the higher temperatures over land and sea in RCM induce larger outbound long wave surface radiation than CPM, by ca. 10 W m$^{-2}$ (Fig. 12f). This, similarly to surface temperate, applies to all analysed composites except one."*

*". Finally, the outbound long wave radiation, similarly to Winter events, shows larger values by RCM, compared to CPM (Fig. 13f)."*

And in the conclusions

*"Regarding differences in surface temperature, RCM showed for most of the analysed EOFs a warmer surface level (by about 1.5 °C). This, in turn, brought larger emissions of outbound long wave radiation in RCM compared to CPM, up to 9 W m$^{-2}$."*

*"Finally, the larger specific humidity north of the Alps in CPM leads to larger CAPE over land, whereas outbound long wave radiation is larger in RCM, linked to the warmer surface level in this set-up."*

*Line 406: there seems to be something missing in this sentence, please check.*

We agree with the editor and have corrected the sentence. The paragraph now reads:

*"The profile and surface humidity and temperature validation has shown that: a) COSMO-CLM performs well in simulating the humidity and temperature lapse-rates, albeit small biases up to 0.2 g kg$^{-1}$ in humidity and 0.5 °C (warm bias) in temperature exist; b) CPM simulates slightly better the vertical humidity profile with a steeper gradient than RCM; c) CPM reduces the positive surface relative humidity bias over locations north of the Alps, e.g., western France, the Czech Republic and eastern Austria."*

*Lines 404 - 407: in the manuscript, please briefly comment on the model compensation errors that reviewer 3 mentions, with respect to the lapse rate.*

This aspect is described between lines L386 and L392 although, we agree, not so clearly. We adapt this lines in the new version of the manuscript to read:

*"The humidity (Fig. 7.c) and temperature (Fig. 7.d) profiles show a wetter mid-troposphere (between 700 hPa and 925 hPa) in RCM than in CPM and a very similar temperature profile between both simulations. CPM simulates slightly better the vertical humidity profile than RCM with a steeper humidity-height gradient. This was also observed in earlier studies with COSMO and COSMO-CLM (Caldas-Alvarez and Khodayar, 2020; Caldas-Alvarez et al., 2021). COSMO-CLM compensates the modelling errors simulating a wetter lower troposphere in RCM to help activate the deep convection parameterization scheme (Tiedtke, 1989). Being of the low-level control type, the Tiedtke deep convection scheme requires a sufficient moisture amount below the cloud base to initiate convection (Doms et al., 2011). By doing so RCM simulates precipitation totals of the same order as CPM that relies more upon the intensification of vertical wind speeds than humidification to simulate convective precipitation. Furthermore, the larger humidity in the mid-troposphere helps reduce the simulated dry-air entrainment increasing the total simulated precipitation. Both simulations show a reliable performance considering the decadal timescales"*

*For the comment from reviewer 3 about Figure 10, the response to reviewers specifies a change to this sentence, however the indicated corrected sentence is not part of the new version of the manuscript. please clarify.*

In our last answers to reviewer 3 we included a sentence from the first submitted version of our manuscript. We included this sentence to show that we aimed to relate the different model variables between one another but that we needed to improve this aspect.

Her/his concern was that "some of the effects illustrated are very closely linked together, e.g. the effects seen in nearsurface specific humidity and surface latent heat flux.". And that this should be pointed out clearly.

In the second submission we conveyed this concern in the detailed description of model variables' differences between RCM and CPM in 7.3 and more generally, in the conclusions. For instance, we related the differences in latent heat fluxes with differences in surface specific humidity and ultimately cape. Likewise, we linked differences in sensible heat fluxes to the differences in surface temperature, affecting in turn OLWR.

We hope to have replied to reviewer 3 and include here again the two related conclusions:

*"For Winter events, latent heat fluxes in CPM were larger over land than in RCM (up to 15 W m-2) on the day prior to severe precipitation. Over the Sea, the opposite occurs, and RCM overestimates the latent heat fluxes compared to CPM (30 W m-2 more). The consequence is an overestimation by CPM of surface specific humidity over land areas north of the Alps compared to RCM (1 g kg-1). However, RCM simulates more specific humidity over the Sea and Italy. The wind transports the moisture excess*

*in RCM inland. Regarding differences in surface temperature, RCM showed for most of the analysed EOFs a warmer surface level (by about 1.5 °C). This, in turn, brought larger emissions of outbound long wave radiation in RCM compared to CPM, up to 9 W m-2."*

*"For Summer events, CPM simulates larger latent heat fluxes over land than RCM, although now restricted to locations north of the Alps. Surface sensible heat fluxes, on the contrary, are larger over land in RCM than in CPM (up to 20 W m-2 more), although these differences are weaker over the Po Valley. The consequence is that CPM simulates larger surface specific humidity north of the Alps whereas RCM simulates larger specific humidity over the Mediterranean and Italy. The different partition of heat fluxes leads to a higher surface temperature in RCM than in CPM over the Alps and northern Europe. Over the Po valley and Italy these differences are weaker or even favourable to CPM. Finally, the larger specific humidity north of the Alps in CPM leads to larger CAPE over land, whereas outbound long wave radiation is larger in RCM, linked to the warmer surface level in this set-up."*

*Additional private note (visible to authors and reviewers only): Overall, the tracked changes version does in many places not correspond to the new version of the manuscript. In the new revision round, please make sure that these versions correspond to each other to allow the reviewers and editors to properly assess your manuscript.*

Yes, we agree that there were changes in the "clean" version in the manuscript, which were not included in the track-changes version. The reason is that the implemented changes were so many, including figures, text, author comments and replies that the track-changes versions became too heavy to work with. We experienced computer problems opening and sharing the track changes version in the latest stages of our revision, so the latest comments and changes had to be included in the "clean" version.

We apologize for any inconvenience and are will obviously make sure to submit coherent versions of the track-changes and "clean" manuscripts in the new round of revisions.

**Author reply to RC1 (wcd-2022-11)**

**General Comment**

*The authors properly addressed my previous comments (and those of the other reviewers) and the manuscript is considerably improved with respect to the former version. I have no further technical objections and in my opinion the paper can be published. However, I still see some weaknesses in the English style, but on this issue I leave the decision to the Editor.*

We thank the reviewer for revising again our manuscript and her/his advice on improving the English style. We will do as suggested and improve the English language.

**Author reply to RC2 (wcd-2022-11)**

We thank the valuable comments and corrections suggested in this second revision.

In the following we reply to all raised issues and include all corrections in the new version of the manuscript.

**Major Comments**

*I do not fully agree with the statement about the overestimation of precipitation over the Alps (L22/L542). For this argument, the authors primarily refer to studies that apparently support their claims. However, the authors of these studies are more careful about that statement, and typically refer to uncertainties related to the gridding procedure, sampling biases due to the gauges being primarily located in valleys, and the prominent under-catch issue by gauges during HP events. I agree with these studies that, while generally useful, the current datasets/observations are not fit for making the "HP overestimation in CPMs" claim in the Alps.*

The statement that our CPM simulation overestimates heavy precipitation intensities is based on the analysis introduced in Sect. 4 and Fig. 5 (empirical PDFs of precipitation). Figure 5 shows that over study region SGer in the period 2000-2015, CPM had some probability  to represent precipitation intensities over 210 mm d$^{-1}$ which is larger than the observed in HYRAS-5km.

The cited publications referred a similar effect in their CPM set ups. Although the differences in the modelling set ups between experiments are numerous, we believe it is worth mentioning that a similar effect has been observed. For instance, Kendon et al., (2012) mention that their 1.5 km decadal simulations in the UK "have a tendency for heavy rain to be too intense". Likewise, Berthou et al., (2018) describe that "mean precipitation is increased over the Alps and becomes larger than in the observations".  Finally, Vergara-Temprado argue that "even at grid spacings of 1 km, convective processes will not be fully resolved […] which might help explaining the overestimation in extreme precipitation intensities at high resolutions".

We acknowledge that we need to improve the description of the uncertainties in such analyses to fairly describe the problem. Hence, we will mention the under-catch issue, sampling problems and gridding procedures that the reviewer and the other publications mention.

This is improved in Sect. 4

*"It should also be noted that even for grid resolutions down to 1 km the updrafts might not me simulated with the right intensity, which can help explain the overestimation of precipitation at these high resolutions (Vergara-Temprado et al., 2020). Also, the comparison against observations must be done carefully as heavy rain measurements might suffer from under catchment, which can reach even 58 % in the worst scenarios (Vergara-Temprado et al., 2020). Furthermore, problems associated with the gridding of precipitation observations and the fact that rain-gauges in the Alpine region tend to be located at the valleys, add uncertainty to the estimation of precipitation and any validation of model data."*

*I am also a bit concerned about the discussion of IWV and its relation to HP (L30 - 35 and L565 – 587). In particular, I did not fully comprehend the presented argument. That is because the authors primarily present results, rather than their idea, and leave most of the interpretation to the reader. It would be helpful if the assessed hypothesis were concisely stated in the introduction, and then assessed in Chapter 8. I currently interpret the results as if the authors alluded to IWV (and remote sensible and*

*latent fluxes over the ocean) being an important driver for differences in HP between RCM and CPM. I don't think that the presented analysis would convincingly outline how such a mechanism would work.*

Thanks for this comment. In our study we were interested in knowing what the differences between RCM and CPM were, regarding simulated variables and processes that affect heavy precipitation. Therefore, we studied model differences of IWV and surface fluxes in the days prior to heavy precipitation.

In the paragraphs mentioned, we present the observed model differences which in general were already present in the seasonal signal. Hence, we cannot attribute the modelling differences in HP to differences in IWV, surface fluxes etc, as they were present in other types of weather regimes. This is why we do not present that mechanism but describe our results which, we believe, could be useful for other modellers that could come across similar differences.

**Minor Comments**

*L75ff: The argument presented by Hohenegger et al. (2009) has meanwhile been augmented and better understood. In particular, the sentence starting on L77 is now outdated. More recent hypothesis actually involve a spatial scale dependency ;-) (e.g., Taylor et al. 2012), that is actually represented in kilometer-resolution climate simulations (e.g., Leutwyler et al. 2021).*

We extend the discussion in the introduction to include the results in these publications

*"Regarding the soil-moisture-precipitation feedback, past work has shown that RCM tends to show a positive sign (Hohenegger et al., 2009; Leutwyler et al., 2021) whereas CPM can show both negative and positive signs at the sub-continental and continental spatial scales, respectively. The reason is that wetter soils induce more frequent precipitation at RCMs but more intense events in CPM with, however, a weak impact on frequency (Leutwyler et al., 2021). CPM seem to better agree with observations as previous observations showed a negative sign of the feedback due to an increased sensible heat flux over drier soils, and mesoscale variability in soil moisture which intensifies afternoon convection (Taylor et al., 2012)."*

*L181/L252: In the revised version of the manuscript the 80th all-day percentile is used, right? Maybe thus mention "all-day" explicitly in the text?*

Yes, we agree. We include it in the text

*L303: The text reads as if the authors are talking about the probability of exceedance. Why not show this metric instead (Figure 5)?*

There are small differences in our analysis as the probabilities shown are empirical. They stand for the number of times a certain precipitation intensity is simulated (or observed) in the data set.

*Section 6/Figure 9: The authors explain what they did, but I do not fully comprehend what they want to demonstrate. For unfamiliar readers (like me), it might be worthwhile to add a sentence or two explaining the intent at the beginning of the Section (same for L440ff), and a few words at the end summarizing the findings.*

We agree. We add the following introductory sentence

*"To understand how differently RCM and CPM represent the main spatial patterns of heavy precipitation we use PCA (Sect. 2.3.2) on events detected in HYRAS-5km in the period 2000-2015. We*

*do this to observe differences in the spatial distributions of heavy precipitation during the most frequent precipitation modes."*

And concluding remark

*"Our analysis shows that RCM and CPM simulate similarly the main precipitation modes up to the fourth principal component in Winter and the third in Summer. These precipitation modes account for 47 % of the precipitation variability in Winter and 37 % in Summer, implying that a large part of the precipitation differences belongs to the secondary modes of precipitation."*

*L485: I do not understand the role of the green shading. Is there something awkward with "negative" sensible heat fluxes? I think the authors need to explain better what the problem is. Also, I am not familiar with the term "surface directed fluxes" used in the caption of Fig 12, and I do not immediately grasp what it means. Please explain it in the caption.*

The problem with negative heat fluxes are the differences between RCM and CPM. If at a certain location RCM shows a large negative heat flux but CPM simulates a weak negative flux, the difference RCM-CPM will be negative. Compared to the other plots, the reader could understand that a negative heat flux stands for a larger emission of heat flux in CPM than in RCM, which is not the case.

We will add a clarification in the text.

*"Figure 12b illustrates these results where differences over the sea are close to zero and green colours denote no positive outbound heat emissions over land. Inbound directed fluxes are dismissed to avoid confusion with the interpretation of the signs in the difference plots. "*

*L497 – L507: I think these paragraphs need to be rewritten. They read like notes rather than actual paragraphs, also I am confused what the underpinning message is.*

We will rewrite them to clearly show the message.

*L572: "overestimates" and "overestimation". I think these words can only be used when comparing against observations. What is wrong with the word "larger"?*

We agree with the reviewer, we correct this in the following version of the manuscript.

*L576: "The wind transports the moisture excess in RCM inland." How do you know that? I think such a statement would require a moisture budget, possibly even including a trajectory analysis.*

We agree that we do not provide quantitative evidence for the moisture transport. We will mention it as a plausible hypothesis explaining the humidity differences observed over Italy.

*"However, RCM simulates more specific humidity over the Sea and Italy, possibly due to the effect of the southerly winds"*

*L595: Maybe the original studies merit citation instead? Chubb et al., (2015) provide a nice summary.*

Yes, thank you for the reference.

**Suggestions**

*Title: "Regional Climate and Convection-Permitting Modelling of heavy precipitation ..." Maybe better: Convection-Parameterizing and Convection-Permitting Simulations of Heavy precipitation […]. The term RCM usually refers to the limited-area extent of the computational domain rather than to the parameterization of convection.*

We agree we adapt the title.

*L97: Maybe add a sentence outlining how the method will be exploited in the presented study?*

Yes, we agree, the following sentence has been added:

*"In this work we will derive composites of relevant model variables and study differences between both modelling set-ups."*

The sentence has been rewritten to

*"In the previous section, we assessed an overestimation of grid-point heavy precipitation for the convection-permitting simulation CPM, but a good performance in detecting severe precipitation events in a 44-year climatology. Here we evaluate the performance of CPM at the event scale validating eight chosen events"*

*L404ff: One might summarize the surface T and QV verification by stating that the event scale verification is consistent with the well-known too dry too hot bias of CPMs. In particular because most of the assessed events are in summer.*

We include this remark.

*L566: I thought you wanted to remove the term "moisture excess" from the manuscript.*

Yes, we correct it in the manuscript.

---

## Author Response (AR3)

**Author reply to the Editor (wcd-2022-11)**

We thank the editor, Prof. Dr. Heinli Wernli for his very helpful comments and indications. We hope we have addressed all remaining concerns, especially those dealing with the English style and composition. The manuscript has been revised by the two senior co-authors as suggested, and the mentioned paragraphs and issues were re-written.

**Editor Comments**

*1) reviewer 2 commented about Section 7.3: "I think these paragraphs need to be rewritten. They read like notes rather than actual paragraphs, also I am confused what the underpinning message is." You changed the sentences a bit but I think the problem remains that this section reads like a collection of statements and does not provide a clear message. After I read all these very short paragraphs I was overwhelmed and confused by the details and did not know what I should get from this. I have the impression that you repeat here too many details that are not so relevant (e.g., low CAPE in winter is maybe clear anyway?). Instead, you should focus on a few key points and combine them to 1 or 2 nice paragraphs. Or you completely delete this section.*

We agree that the composition of this section was not clear. It has been fully rewritten, reducing the extension from 1.5 pages to 0.5 pages and the last figure (former Fig. 13) has been removed. This has been done to avoid redundant information and unnecessary details. We hope it reads more clearly now. Please refer to the track changes version to see the differences.

*2) Please have again a very careful look at your English (find a senior colleague who has time to work through the text once again). For instance, in L35 "Regarding surface temperature RCM simulates up to 2 °C more than CPM ..." does not sound very nice, please improve. Why not just "Surface temperatures in RCM are up to 2 degC higher than in CPM"? Similarly on L532. And I don't understand in L14 "not all implications of reaching CPM are known" ... There are many sentences that could/should be written a bit more elegantly and clearly. I emphasise this because it was already mentioned twice by the reviewers.*

We have revised the style and the English language of the text. Please refer to the track changes version to see the differences.

*3) Please also carefully check formatting issues: units should not be in italics (e.g., L13), avoid typos (L23), exponents should be properly formatted (L497 and in other places), "Winter" should not be capitalised (L543), etc.*

We have corrected these issues.

*4) The huge precipitation differences you mention in L28, are they for a particular day or in the climatology, this is unclear.*

They refer to specific heavy precipitation events. It has been rephrased in the new version of the manuscript.

*5) L29: The sentence "Either RCM or CPM can show these large differences ..." does not make sense because one model alone does not produce "differences"*

We agree with the comment. It has been rephrased.

---

## Author Response (AR4)

**Author reply to the Editor (wcd-2022-11)**

We are very happy and honoured to get our work accepted at WCD and are hoping to publish again with WCD and Copernicus in the future. We thank Prof. Dr. Heinli Wernli, and Prof. Dr. Daniela Domeisen for acting as editors, the three independent referees and the people at Copernicus, involved in the revision of our manuscript for their valuable time and work. Especially, since this review process has been longer than one year. We are very grateful for the patience and understanding regarding the needed deadline extensions and are very satisfied with the final outcome and conclusions of the paper.

In the following we provide our last comments and changes

**Editor Comments**

*In the captions of Figs. 3 and 4, it would be good to mention whether you consider here the 98th percentile in the region SGer or CPM; just add "... for the 98-percentile most severe precipitation case in the region XXX ..."*

It is region SGer. It has been included

*You deleted the misleading sentence ""Either RCM or CPM can show these large differences ..." but there is a similarly strange formulation still in the conclusions on L535: "Composite maps derived from the leading modes showed that either RCM or CPM can represent daily precipitation differences as large as 200 m d-1, although CPM tends to simulate larger precipitation than RCM over the mountains." Again, "either A or B can represent differences" is not very meaningful. And also the unit should most likely be mm/d, not m/d. Maybe best to also delete this entire sentence? Or phrase in a better way?*

We replace that sentence for another phrase, similar to what is said in the abstract.

*"Composite maps derived from the leading modes showed that CPM systematically represents more precipitation at the mountain tops, but that RCM may show large intensities (up to 200 mm d$^{-1}$) in other regions."*